Manuscript prepared for The Cryosphere
with version 2015/04/24 7.83 Copernicus papers of the LATEX class copernicus.cls.
Date: 25 January 2017

# Signature of Arctic first-year ice melt pond fraction in X-band SAR imagery

Ane S. Fors[1], Dmitry V. Divine[2,3], Anthony P. Doulgeris[1], Angelika H. H. Renner[2,4], and Sebastian Gerland[2]

[1]Department of Physics and Technology, University of Tromsø - The Arctic University of Norway, 9037 Tromsø, Norway
[2]Norwegian Polar Institute, FRAM Centre, 9296 Tromsø, Norway
[3]Department of Mathematics and Statistics, University of Tromsø - The Arctic University of Norway, 9037 Tromsø, Norway
[4]Institute of Marine Research, 9294 Tromsø, Norway

*Correspondence to:* Ane Fors (ane.s.fors@uit.no)

**Abstract.** In this paper we investigate the potential of melt pond fraction retrieval from X-band polarimetric synthetic aperture radar (SAR) on drifting first-year sea ice. Melt pond fractions retrieved from a helicopter-borne camera system were compared to polarimetric features extracted from four dual polarimetric X-band SAR scenes, revealing significant relationships. The correlations were strongly dependent on wind speed and SAR incidence angle. Co-polarisation ratio was found to be the most promising SAR feature for melt pond fraction estimation at intermediate wind speeds (6.2 m/s), with a Spearman's correlation coefficient of $0.46$. At low wind speeds (0.6 m/s), this relation disappeared due to low backscatter from the melt ponds, and backscatter VV-polarisation intensity had the strongest relationship to melt pond fraction with a correlation coefficient of $-0.53$. To further investigate these relations, regression fits were made both for the intermediate ($R^2_{fit} = 0.21$) and low ($R^2_{fit} = 0.26$) wind case, and the fits were tested on the satellite scenes in the study. The regression fits gave good estimates of mean melt pond fraction for the full satellite scenes, with less than $4\%$ from a similar statistics derived from analysis of low altitude imagery captured during helicopter ice-survey flights in the study area. A smoothing window of $51 \times 51$ pixels gave the best reproduction of the width of the melt pond fraction distribution. A considerable part of the backscatter signal was below the noise floor at SAR incidence angles above $\sim 40°$, restricting the information gain from polarimetric features above this threshold. Compared to previous studies in C-band, limitations concerning wind speed and noise floor set stricter constraints on melt pond fraction retrieval in X-band. Despite this, our findings suggest new possibilities in melt pond fraction estimation from X-band SAR, opening for expanded monitoring of melt ponds during melt season in the future.

# 1 Introduction

Melt ponds form from snow and ice melt water on the Arctic sea ice during spring and summer, and can cover up to $50 - 60\%$ of the sea ice surface (Perovich, 2002; Eicken et al., 2004; Inoue et al., 2008; Perovich et al., 2009; Polashenski et al., 2012). Their presence affects the heat budget of the sea ice by decreasing the surface albedo, which increases the solar absorption in the ice volume and the transmission of solar energy to the ocean (Eicken et al., 2004; Ehn et al., 2011; Nicolaus et al., 2012; Perovich and Polashenski, 2012). The transmission is generally larger for first-year ice (FYI) than for multiyear ice (MYI) due to FYI's lower sea ice thickness. (Light et al., 2008; Nicolaus et al., 2012; Hudson et al., 2013). FYI also often experiences higher melt pond fractions ($f_{MP}$) than MYI (Fetterer and Untersteiner, 1998; Nicolaus et al., 2012). The increased absorption induced by melt ponds accelerates the decay of sea ice, and the intensified warming of the ocean possibly delays the ice growth in the autumn (Flocco et al., 2012; Holland et al., 2012; Hudson et al., 2013; Schröder et al., 2014; Flocco et al., 2015). Formation and evolution of melt ponds are poorly represented in sea ice models, potentially contributing to an underestimation of the observed sea ice extent reduction in model projections (Flocco et al., 2012; Holland et al., 2012; Flocco et al., 2015). An increased number of observations of $f_{MP}$ for different sea ice types at regional scale is needed to improve the understanding of the role of melt ponds in the Arctic climate system. Satellite imagery offers good opportunities for such large scale monitoring of melt ponds.

Several algorithms have been developed for retrieval of $f_{MP}$ from optical satellites, measuring the spectral reflectance from open water, sea ice and melt ponds. The algorithms apply to different multispectral sensors; the enhanced thematic mapper plus (ETM+) on board Landsat 7 (Markus et al., 2003; Rösel and Kaleschke, 2011), moderate-resolution imaging spectroradiometer (MODIS) on board Aqua and Terra (Tschudi et al., 2008; Rösel et al., 2012; Rösel and Kaleschke, 2012), and medium resolution imaging spectrometer (MERIS) on board Envisat (Zege et al., 2015; Istomina et al., 2015). Commonly, the retrieval algorithms are vulnerable to correction for atmospheric constituents and influences of the viewing angles and the solar geometry. They also require cloud-free conditions, limiting their applicability in the Arctic due to the persistent cloud cover present during summer. Satellite microwave radiometers and scatterometers can on the other hand penetrate clouds, but their resolution is in general too coarse for automated melt pond monitoring (Comiso and Kwok, 1996; Howell et al., 2006).

Satellite synthetic aperture radar (SAR) offers independence of cloud cover, atmospheric constituents, and daylight, combined with high spatial resolution. Several studies have focused on $f_{MP}$ retrieval from single polarisation SAR, transmitting and receiving either vertical (VV) or horizontal (HH) polarised waves. Jeffries et al. (1997) developed a model for $f_{MP}$ retrieval over MYI floes in the Beaufort sea from ERS-1 SAR satellite images, but lack of wind consideration limit the validity of the model. Wind speed was found to be a key parameter when Yackel and Barber (2000) demonstrated a significant relation between $f_{MP}$ and HH intensity on land-fast FYI within the Cana-

dian Arctic Archipelago using SAR satellite scenes from Radarsat-1. The relationship was strong at intermediate wind speeds, but lacking at low wind speeds. Mäkynen et al. (2014) compared $f_{MP}$
retrieved from MODIS and from a large amount of ENVISAT ASAR satellite scenes. The study area covered both FYI and MYI north of the Fram Strait. The study concluded that $f_{MP}$ estimation was not possible based on the investigated data set. The above-mentioned studies all focus on C-band frequency ( 5.4 GHz) SAR. Kern et al. (2010) investigated the use of supplementary frequencies for $f_{MP}$ retrieval on MYI in the Arctic Ocean, and showed promising results in combining C, Ku
( 17.2 GHz) and X ( 9.6 GHz) band data from a helicopter-borne scatterometer. Estimation of $f_{MP}$ in X-band satellite SAR was further explored by Kim et al. (2013), investigating melt ponds in a TerraSAR-X scene acquired over MYI in the Chukchi Sea. Only large melt ponds were found detectable in the study, leading to an underestimation of $f_{MP}$. All in all, retrieval of $f_{MP}$ from single polarimetric SAR has proven to be difficult.

Dual and quad polarimetric SAR transmit and receive both vertical and horizontal waves, resulting in four possible channel combinations (HH, HV, VH and VV), and give information about the polarisation properties of the backscatter in addition to single channel intensity variations. The channels can be combined into polarimetric SAR features, e.g. channel ratios, reducing the dependency of sensor geometry. Based on C-band scatterometer measurements, Scharien et al. (2012) suggested
co-polarisation ratio ($R_{VV/HH}$) to give an unambiguous estimation of $f_{MP}$ at large incidence angles for land-fast FYI in the Canadian Arctic Archipelago and the Beaufort Sea. The topic was further investigated (Scharien et al., 2014b), and expanded to Radarsat-2 satellite scenes in Scharien et al. (2014a), demonstrating a strong potential of $f_{MP}$ estimation from C-band dual polarimetric space-borne SAR. Both studies were performed in the central Canadian Arctic Archipelago. The
findings were partly confirmed by Fors et al. (2015), who also suggest a relationship between $f_{MP}$ and the statistical SAR feature relative kurtosis ($RK$) utilizing Radarsat-2 on iceberg-fast FYI and MYI in the Fram Strait. Han et al. (2016) combined multiple polarimetric SAR features in $f_{MP}$ estimation by machine learning methods, employing the co-polarisation channels of the MYI X-band SAR scene explored in Kim et al. (2013). An additional scene was also included in the study, though
without melt pond information. The study showed promising results, but the authors claim that more scenes with various sea ice types and incidence angles are needed to develop a general propose $f_{MP}$ model. Lack of wind information is also limiting the relevance of the study.

In summary, the main achievements on $f_{MP}$ retrieval with SAR come from *dual polarimetric* C-band studies on land-fast FYI. The potential of $f_{MP}$ retrieval with polarimetric X-band SAR has
only been explored in one single study by Han et al. (2016), focusing on MYI. Hence, there is a need for more studies on the influence of $f_{MP}$ on polarimetric X-band SAR imagery. As MYI and land-fast FYI have been the main focus in previous studies, there is also a need to expand to other sea ice types. Drifting FYI is becoming more prominent in the Arctic with the recent shift to a thinner, more

seasonal, and more mobile sea ice cover (Perovich et al., 2015), and the polarimetric SAR signature

of $f_{MP}$ in drifting FYI needs more attention.

The objective of this study is to investigate the potential of $f_{MP}$ retrieval from level drifting FYI with dual-polarisation X-band satellite SAR. A data set consisting of four high resolution dual-polarisation TerraSAR-X satellite scenes, combined with $f_{MP}$ retrieved from a helicopter-borne camera system forms the basis of the study. TerraSAR-X offers very high resolution dual-

polarimetric images, with a strong sensitivity to micro-scale surface roughness due to the high frequency. Both the high resolution and sensitivity to surface roughness can be advantages in $f_{MP}$ investigations. The data were collected north of Svalbard in summer 2012. We explore the correlation between $f_{MP}$ and different polarimetric SAR features extracted from the HH and VV channels. Based on the results, we suggest two simple empirical regression fits for $f_{MP}$ estimation adjusted to

an intermediate and a low-wind speed case. The influence and limitations related to wind conditions, incidence angle, noise floor, scale and surface roughness are discussed in light of the results.

## 2 Melt ponds in SAR imagery

The signature of melt ponds in SAR imagery depends on both melt pond properties and radar parameters. Wind at the sea ice surface changes the surface roughness of the melt ponds, and hence

their SAR backscatter signature and contrast to the surrounding sea ice. The influence of wind is dependent on fetch length, depth of the ponds, orientation of the ponds and the topography of the surrounding sea ice (Scharien et al., 2012, 2014b). During very calm conditions, the scattering from melt ponds is mainly specular. This occurs at wind speeds of $2-3$ m/s in 10 m height ($U_{10}$) in C-band, in agreement with findings for ocean surfaces ($\sim 2.0$ m/s at $0°C$) (Donelan and Pierson,

1987; Scharien et al., 2012, 2014b). A similar threshold in X-band equals $\sim 2.8$ m/s (Donelan and Pierson, 1987). Refrozen ponds suppress the wind wave surface roughness induced on open ponds, and yield a signature closer to newly formed sea ice (Yackel et al., 2007; Scharien et al., 2014b, a). The size distribution of melt ponds also affects their SAR signature. Ponds smaller than the SAR resolution return a signal mixed with sea ice and possibly leads, while very large melt ponds could

fill a resolution cell. Choice of SAR resolution and speckle smoothing window size could hence affect the SAR $f_{MP}$ signature. The coverage of melt ponds varies during the melt season, increasing from melt onset until it reaches a maximum level, and then gradually reducing as the ponds starts to drain (Barber et al., 2001). At the end of the melt season, the remaining melt ponds refreeze.

The SAR signature of melt ponds changes with incidence angle of the satellite. Scharien et al.

(2012) found a larger decrease in C-band SAR intensity ($\sigma^0$) with increasing incidence angle for melt ponds than for sea ice. In contrast to sea ice, $\sigma^0_{HH}$ decreased more than $\sigma^0_{VV}$ for melt ponds. The most suitable incidence angle ranges for $f_{MP}$ retrieval is method dependent. SAR frequency also influences the melt pond signature (Kern et al., 2010). X-band is more sensitive to small-scale

surface roughness than C-band, as the effect of surface roughness depends on radar wavelength. In addition, the sea ice volume penetration depth decreases with increasing frequency, leading to less volume scattering from sea ice at higher frequencies.

Several dual-polarimetric SAR features have been suggested for $f_{MP}$ retrieval from SAR, utilizing different expected relations to physical properties of sea ice and melt ponds (Scharien et al., 2012, 2014a; Fors et al., 2015; Han et al., 2016). Eight of these features are included in our study and are described in the following subsection.

## 2.1 Polarimetric SAR features

For a fully polarimetric SAR system, which transmits and receives both horizontally (H) and vertically (V) polarised waves, the scattering matrix can be written as

$$\boldsymbol{S} = \begin{bmatrix} S_{HH} & S_{VH} \\ S_{HV} & S_{VV} \end{bmatrix} = \begin{bmatrix} |S_{HH}|e^{j\phi_{HH}} & |S_{VH}|e^{j\phi_{VH}} \\ |S_{HV}|e^{j\phi_{HV}} & |S_{VV|}e^{j\phi_{VV}} \end{bmatrix}, \tag{1}$$

where $|\cdot|$ and $\phi_{xx}$ denote the amplitude and the phase of the measured complex scattering coefficients, respectively (Lee and Pottier, 2009). Assuming reciprocity ($S_{HV} = S_{VH}$), the Pauli basis scattering vector, $\boldsymbol{k}$, can be extracted from $\boldsymbol{S}$ as

$$\boldsymbol{k} = \frac{1}{\sqrt{2}} \begin{bmatrix} S_{HH} + S_{VV} & S_{HH} - S_{VV} & 2S_{HV} \end{bmatrix}^{\dagger}, \tag{2}$$

where $\dagger$ denotes the transpose operator (Lee and Pottier, 2009). In our study, we are only utilizing the co-polarisation channels (HH and VV), and so the scattering vector reduceds to

$$\boldsymbol{k} = \frac{1}{\sqrt{2}} \begin{bmatrix} S_{HH} + S_{VV} & S_{HH} - S_{VV} \end{bmatrix}^{\dagger}. \tag{3}$$

The sample coherency matrix, $\boldsymbol{T}$, is defined as the mean Hermitian outer product of the Pauli basis scattering vector:

$$\boldsymbol{T} = \frac{1}{L} \sum_{i=1}^{L} \boldsymbol{k}_i \boldsymbol{k}_i^{*\dagger}, \tag{4}$$

where $\boldsymbol{k}_i$ is the single-look complex vector corresponding to pixel $i$, $L$ is the number of scattering vectors in a local neighborhood, and $*$ denotes the complex conjugate (Lee and Pottier, 2009). Similarly, in the dual-polarisation case, the Lexicographic basis scattering vector, $\boldsymbol{s}$, can be written as

$$\boldsymbol{s} = \begin{bmatrix} S_{HH} & S_{VV} \end{bmatrix}^{\dagger}. \tag{5}$$

Based on $\boldsymbol{s}$ , the sample covariance matrix, $\boldsymbol{C}$, is defined as

$$\boldsymbol{C} = \frac{1}{L} \sum_{i=1}^{L} \boldsymbol{s}_i \boldsymbol{s}_i^{*\dagger}, \tag{6}$$

where $\boldsymbol{s}_i$ is the single look complex vector corresponding to pixel $i$ (Lee and Pottier, 2009).

The SAR intensity ($\sigma^0$) is retrieved from a single polarisation channel, defined by the amplitudes of the complex scattering coefficients,

$$\sigma^0_{VV} = \langle |S_{VV}|^2 \rangle \text{ and } \sigma^0_{HH} = \langle |S_{HH}|^2 \rangle, \tag{7}$$

were $\langle \cdot \rangle$ denotes an ensemble average. The relation between these basic features and $f_{MP}$ have been investigated in several studies (Jeffries et al., 1997; Yackel and Barber, 2000; Mäkynen et al., 2014; Kern et al., 2010; Kim et al., 2013). However, carrying information from one single polarisation channel only, makes them less robust than polarimetric features that hold information from several channels.

Co-polarisation ratio ($R_{VV/HH}$) has so far been the most promising SAR feature for $f_{MP}$ extraction in C-band (Scharien et al., 2014a). It is defined as the ratio between the intensities of the co-polarisation complex scattering coefficients

$$R_{VV/HH} = \frac{\langle |S_{VV}|^2 \rangle}{\langle |S_{HH}|^2 \rangle}. \tag{8}$$

For smooth surfaces within the Bragg scatter validity region, $R_{VV/HH}$ depends only on the surface complex permittivity and local incidence angle, and is independent of surface roughness (Hajnsek et al., 2003). Both freshwater and saline melt ponds have considerably higher complex permittivity than sea ice, and $R_{VV/HH}$ has therefore been suggested for $f_{MP}$ retrieval (Scharien et al., 2012, 2014b, a). The Bragg criterion is fulfilled for $ks_{RMS} < 0.3$, where $k$ is the wavenumber and $s_{RMS}$ is the root mean square height of the sea ice surface, describing its surface roughness. This corresponds to $s_{RMS} < 2.8$ mm in C-band, and $s_{RMS} < 1.4$ mm in X-band. The sea ice surface roughness was found too high to fill the criterion in studies north of Spitsbergen and in the Fram Strait (Beckers et al., 2015; Fors et al., 2016b), while Scharien et al. (2014b) found land-fast ice in the central Canadian Arctic Archipelago to fulfill the criterion at C-band, and partly at X-band. In the same study, melt ponds filled the criterion at wind speeds below 6.4 m/s in C-band, corresponding to $\sim 5.5$ m/s in X-band (Scharien et al., 2014b). When the Bragg criterion is exceeded, $R_{VV/HH}$ decreases with increasing surface roughness. $R_{VV/HH}$ increases with incidence angle, and Scharien et al. (2012) found incidence angles above 35° to be most appropriate for $f_{MP}$ retrieval based on $R_{VV/HH}$ in C-band.

Relative kurtosis ($RK$) is a statistical measure of non-Gaussianity, which describes the shape of the distribution of scattering coefficients in SAR scenes. It has previously been used for sea ice segmentation (Moen et al., 2013; Fors et al., 2016a). It is defined as Mardia's multivariate kurtosis of a sample, divided by the expected multivariate kurtosis of a complex normal distribution

$$\text{RK} = \frac{1}{L} \frac{1}{d(d+1)} \sum_{i=1}^{L} \left[ \mathbf{s}_i^{*\dagger} \mathbf{C}^{-1} \mathbf{s}_i \right]^2, \tag{9}$$

where $d$ is the number of polarimetric channels (Mardia, 1970; Doulgeris and Eltoft, 2010). It has a potential in $f_{MP}$ retrieval as it is sensitive to mixtures of surfaces. At C-band, $RK$ was found significantly correlated to $f_{MP}$ over iceberg-fast sea ice in the Fram Strait (Fors et al., 2015).

Entropy ($H$) is a part of the $H/A/\overline{\alpha}$ polarimetric decomposition, based on the eigenvectors and eigenvalues of $\boldsymbol{T}$, describing SAR scattering mechanisms (Cloude and Pottier, 1997). $H$ is a measure of the randomness of the scattering processes, and is defined as

$$H = -\sum_{i=1}^{d} p_i \log_d p_i, \tag{10}$$

where $p_i$ is the relative magnitude of each eigenvalue

$$p_i = \frac{\lambda_i}{\sum_{k=1}^{d} \lambda_k}, \tag{11}$$

and $\lambda_i$ is the $i^{th}$ eigenvalue of $\boldsymbol{T}$ ($\lambda_1 > \lambda_2$) (Cloude and Pottier, 1997). Only the co-polarisation channels (HH and VV) are included in our study ($d = 2$), and a dual polarisation version of the entropy, denoted $H'$, is therefore used (Cloude, 2007; Skrunes et al., 2014). $H' = 0$ indicates a single dominant scattering mechanism, while $H' = 1$ indicates a depolarized signal. In the case of dual polarisation, $H'$ and anisotropy represent the same information as they both only depends on $\lambda_1$ and $\lambda_2$, and anisotropy is therefore not included in our study.

The alpha angle of the largest eigenvalue ($\alpha_1'$) describes the type of the dominating scattering mechanism. It is expressed as

$$\alpha_1' = \cos^{-1} \frac{|x_1|}{|v_1|}, \tag{12}$$

where $x_1$ is the first element of the largest eigenvector, and $|v_1|$ is the norm of the first eigenvector (Lee and Pottier, 2009). The feature can be written as a function of $R_{VV/HH}$ for slightly rough surfaces, and will then increase with increasing complex permittivity (van Zyl and Kim, 2011).

Co-polarisation correlation magnitude ($|\rho|$) is defined as

$$|\rho| = \left| \frac{\langle S_{\text{HH}} S_{\text{VV}}^* \rangle}{\sqrt{\langle S_{\text{HH}} S_{\text{HH}}^* \rangle \langle S_{\text{VV}} S_{\text{VV}}^* \rangle}} \right|, \tag{13}$$

and describes the degree of correlation between the co-polarisation channels (Drinkwater et al., 1992). A perfect correlation returns unity, while depolarisation of the signal will reduce the magnitude. Complex surfaces, multiple scattering surface layers and/or presence of system noise could depolarize the signal (Drinkwater et al., 1992).

Phase difference ($\angle \rho$) is expressed as (Drinkwater et al., 1992)

$$\angle \rho = \angle \left( \langle S_{\text{HH}} S_{\text{VV}}^* \rangle \right). \tag{14}$$

As the relative phase of the co-polarisation waves is changed in every scattering event, the mean and standard deviation of $\angle \rho$ are related to the scattering history (Eom and Boerner, 1991; Drinkwater

et al., 1992). Han et al. (2016) found $H$, $\alpha_1'$, $|\rho|$, and $\angle\rho$ to give useful information for $f_{MP}$ retrieval at X-band.

## 3 Methods

### 3.1 Study region and sea ice conditions

The ICE2012 campaign took place on drifting FYI north of Svalbard, in the southwestern Nansen Basin (Fig. 1), where the research vessel R/V Lance was moored up to an ice floe for eight days. The sea ice cover in the area is generally dominated by first- or second-year ice with only moderate amounts of deformation (Renner et al., 2013). While large seasonal variability exists in the area, summer ice thickness has been fairly stable since 2007. However, Renner et al. (2013) found further

indicators for a trend towards younger sea ice in the region. Little deformation and dominance of young ice leads to relatively low sea ice surface roughness, with a root mean square height of around or less than 0.1 m in the region (Beckers et al., 2015). Substantial snow cover can accumulate during spring, however, during the summer season, the snow melts completely contributing to extensive melt pond formation.

During the ICE2012 campaign, regular sea ice thickness and melt pond surveys were performed on the ice and from helicopter. Modal ice thickness in the region was less than in previous years with 0.7 to 0.9 m (Divine et al., 2015). The very close drift ice was fairly level with less than 10% deformed ice. Sea ice surface roughness retrieved from the floe by R/V Lance is given in Table 3. The surface roughness values are expected to be representative for the whole study region, as the sea

ice in the area was found to be very uniform (Hudson et al., 2013; Divine et al., 2015). The values also agree well with values derived from laser altimeter observations by Beckers et al. (2015).

At the time of the campaign, all snow had melted and extensive networks of melt ponds led to an average $f_{MP}$ of 26% of the sea ice area (Divine et al., 2015). The melt ponds were mostly whithin 15 to 30 cm deep, however, extensive melt led to some ponds having melted through the ice slab.

The water in the pond networks was therefore mostly saline.

Hudson et al. (2013) report an average thinning of the sea ice next to R/V Lance of over 17 cm between 28 July and 2 August which to a large degree can be explained by absorption of atmospheric and oceanic heat by the ice. Air temperatures were varied little between $-1$ to $1.5^\circ C$. Combined with the oceanic heat flux, the ice was therefore in continuous melt even at nighttime. Meteorological

conditions were dominated by heavy cloud cover with only short spells of incomplete or thin cloud cover. Ice cores were taken every other day between 27 July and 2 August with an additional core on 28 July for chemical analysis. They confirm the presence of a consistent 4 to 5 cm thick surface scattering layer of white, granular, deteriorated ice. Temperature profiles through the ice were fairly stable with vertical variations between near $0^\circ C$ at the surface to $-1$ to $-1.3^\circ C$ at the bottom.

Salinity measurements show very low values in the upper 20 cm with salinities of less than 1 psu

and increasing to 3 to 4 psu near the bottom, in agreement with the advanced stage of melt of the ice cover.

### 3.2 Data set

*In situ* and helicopter-borne measurements from ICE2012 are combined with four high-resolution TerraSAR-X (TS-X) satellite scenes. The satellite scenes are StripMap mode acqusitions, with a HH-VV channel combination (see Table 1 and Fig. 1). The scene labeled T1 was acquired in descending orbit, while T2-T4 were acquired in ascending orbits. All scenes were converted to ground range and radiometrically calibrated to $\sigma^0$. The noise equivalent $\sigma^0$ (NESZ) was then subtracted. The absolute radiometric calibration accuracy of TSX is 0.6 dB (Airbus Defence and Space, 2013). For comparison with $f_{MP}$ retrieved from helicopter-borne data, the scenes were geocoded with ESA's Sentinel-1 toolbox, SNAP (European Space Agency, 2016). All analysis were, however, performed in SLC range and azimuth coordinates. Open water areas were not included in our study. For each satellite scene, these areas were masked out with a simple binary mask. The mask was created by filtering the scenes with a $13 \times 13$ pixels averaging sliding window, and manually setting a lower sea ice threshold value on $\sigma^0_{HH}$ in each scene (-18 dB,-17 dB,-16 dB and -18 dB, for T1-T4 respectively). Regions with less than 750 pixels ($\sim 5000 \ m^2$) were merged into the surrounding region (open water or sea ice) to smooth the mask.

A stereocamera system (ICE stereocamera system) was mounted in a single enclosure outside the helicopter during ICE2012 (Divine et al., 2016). The system consisted of two cameras (Canon 5D Mark II), combined with GPS/INS (Novatel) and a laser altimeter. $f_{MP}$ was retrieved from downward-looking images captured by one of the cameras during five helicopter surveys performed between 31 July and 2 August 2012 (see Table 2 and Fig. 1). The footprint of the images was about $60 \times 40$ m for a typical flight altitude of about 35 m, and the images were not overlapping. A full description of the method is given in Divine et al. (2015). In our study, $f_{MP}$ was calculated from the processed images without sea water fraction ($\sim 5700$ images), to better match the sea ice mask. This excluded $f_{MP}$ from the ice edges and small floes, resulting in a slightly higher $f_{MP}$ than that obtained in Divine et al. (2015).

The ICE stereocamera system was also used to investigate sea ice surface topography at the floe where R/V Lance was anchored. For this purpose, the cameras shot sequentially with a frequency of 1 Hz to ensure sufficient overlap between subsequent images during the flights. Using photogrammetric technique, the sequences of overlapping images were used to construct a digital terrain model (DTM) of the sea ice surface. DTMs were generated for five selected segments of the ICE12 ice floe with a spatial resolution of 2 cm. Surface roughness, in form of root mean square height of the sea ice surface ($s_{RMS}$), was estimated from the DTMs using random sampling to account for spatial auto-correlation. Only grid nodes above the water level were used. The accuracy of the retrieved

$s_{RMS}$ were $\pm 4$ cm according to *in situ* measurements from two test areas. A full description of the method is given in Divine et al. (2016).

An automatic weather station located at the floe where R/V Lance was moored during ICE2012 measured wind speed and air temperature 2 m above the sea ice surface(Hudson et al., 2013). Wind
speed ($U_2$) was measured with a three-dimensional ultrasonic anemometer (Campbell Scientific Inc., CSAT3), and air temperature was measured with a temperature probe (Vaisala, HMP155) in an unventilated radiation shield. Tab. 1 presents air temperature and 10 minutes averaged wind speed at the time of the satellite acquisitions.

### 3.3 Design of study

An easy recognizable sea ice floe present in two of the investigated satellite scenes (T3 and T4) is the main focus of our study (see Fig. 2). This floe was chosen as it allowed for a reliable co-location between airborne images and satellite scenes, and was present in more than one scene. The rest of the airborne track was not possible to co-locate exact enough for a high-quality study. The floe had a diameter of $\sim 3.6$ km, and a collection of 43 images was captured across the floe during the $2^{nd}$
helicopter flight on 2 August 2012 (see Tab. 2). The time offset between the flight and acquision of T4 was $\sim 40$ minutes. The position of the helicopter images had to be corrected for sea ice drift to retrieve co-location between the images and the floe captured in T4. As a first step, the image center coordinates were shifted according to drift information from GPS tracks of R/V Lance, positioned $\sim 25$ km south of the floe at the time of acquisition. Second, the track was manually adjusted by
fitting the helicopter images with ground features, such as ice edges and areas with open water. Co-location of the helicopter images and the floe in T3 was based on the one of T4. The maximum error of the co-location was estimated to be 7 m lengthwise and crosswise the flight direction, resulting in a maximum possible areal offset of 27% between the satellite scene and each helicopter image. After co-location, mean and standard deviation of the polarimetric SAR features were calculated for
the pixels underlying each of the helicopter images.

The statistical dependence between the extracted SAR features and the corresponding $f_{MP}$ retrieved from each of the 43 helicopter images was evaluated with the non-parametric Spearman's rank correlation coefficient ($r$). For a sample size of $n$ images, $r$ is defined as

$$r = 1 - \frac{6 \sum d_i^2}{n(n^2 - 1)}, \tag{15}$$

where $d_i$ is the difference in paired rank number $i$ (Corder and Foreman, 2009). Rank ties are assigned a rank equal to the average of their position in the ascending order of the values. The coefficient takes values between -1 and 1, where values of $\pm 1$ correspond to full correlation, while 0 corresponds to no correlation. A negative sign indicates an inverse relationship. Spearman's correlation coefficient assumes a monotonic relationship. It is used instead of Pearson's linear correlation

coefficient, to allow for non-linear correlations. It is also less sensitive to outliers than Pearson's correlation coefficient. Correlations were considered significant if they had p-values below 0.05.

Two regression fits were proposed from the correlation results, representing an intermediate and a low-wind case. A least squares linear fit with bisquare weights was used to construct the regression fits (Hoaglin et al., 1983). The regression fits were applied to the full area of the floe in T3 and T4,

and to the full area of the four satellite scenes included in the study (T1-T4). The estimated $f_{MP}$ distributions were compared and evaluated towards the observed $f_{MP}$ distribution retrieved from all the helicopter flights included in the study (see bottom entries Tab. 2). The effect of smoothing was tested by using a range of different averaging sliding smoothing window sizes ($13 \times 13$ to $51 \times 51$ pixels) in the $f_{MP}$ estimation. Incidence angle correction was applied to the scenes for a better

comparison, employing the following equation (Kellndorfer et al., 1998)

$$\sigma^0_{corr} = \sigma^0 \frac{\sin(\theta)}{\sin(\theta_{ref})}, \tag{16}$$

where $\sigma^0$ is the original backscatter coefficient, $\theta$ is the center incidence angle of the scene to be corrected, and $\theta_{ref}$ is the reference incidence angle of scene T4. The correction was only applied in the low-wind case, as it canceled in the intermediate wind case due to the use of a co-polarisation

ratio.

## 4 Results

This section presents the results of the correlation analysis examining the relation between the investigated polarimetric SAR features and observed $f_{MP}$. It then present a brief signal-to-noise analysis, before it focuses on $f_{MP}$ retrieval in an intermediate and a low-wind case.

### 4.1 Correlation between polarimetric SAR features and $f_{MP}$

Correlation coefficients ($r$) between $f_{MP}$ retrieved from the 43 helicopter images of the investigated floe, and the mean and standard deviation of the polarimetric SAR features extracted from the corresponding areas in scenes T3 and T4, are presented in Table 4. Values significant within a $95\%$ confidence interval are highlighted in bold. In scene T3, $R_{VV/HH}$ shows the strongest correlation to

$f_{MP}$. In addition, the mean of $\alpha_1$ is significantly correlated to $f_{MP}$. None of the other investigated SAR features are significantly correlated to $f_{MP}$ in scene T3. In scene T4, the mean values of $\sigma^0_{HH}$, $\sigma^0_{VV}$ and $R_{VV/HH}$ are significantly correlated to $f_{MP}$, the strongest correlation is found for $\sigma^0_{VV}$. Some of the standard deviation values are also correlated to $f_{MP}$. In scene T4, NESZ subtraction had large influence on the results indicating that the signal is close to, or reaching the noise floor.

Figure 3 confirms the low signal-to-noise ratio in T4. We show the 10, 25, 50, 75 and 90 percentiles of $\sigma^0_{HH}$ (dB) and $\sigma^0_{VV}$ (dB) retrieved for four different $f_{MP}$ intervals on the floe present in scene T3 (top) and T4 (bottom), combined with the noise floor of the HH and VV channels. In T3, less than

10% of the signal is below the noise floor ($\sim -25$ dB). Both $\sigma^0_{HH}$ and $\sigma^0_{VV}$ are increasing with $f_{MP}$. $\sigma^0_{VV}$ has the steepest increase, confirming an increase in $R_{VV/HH}$ with $f_{MP}$ (Tab. 4). In scene T4, the backscatter signal is weaker and noise floor is higher than in scene T3 ($\sim -21$ dB), both due to the higher incidence angle of scene T4 (see Tab. 1). This brings as much as $25\%$ of the signal below the noise floor. The strength of the signal decreases with $f_{MP}$, implying specular reflection from the melt ponds, supported by the low wind speed (0.6 m/s) at acquisition of scene T4 (see Tab. 1). The difference between $\sigma^0_{HH}$ and $\sigma^0_{VV}$ is decreasing with $f_{MP}$, confirming an inverse relation between $R_{VV/HH}$ and $f_{MP}$ in T4 (Tab. 4). In scene T1 and T2, the noise floors are $\sim 23$ dB, leaving $\sim 15\%$ of the signal below the noise floor.

The melt ponds affect the polarimetric signatures in scene T3 and T4 differently (Table 4 and Fig. 3), mainly due to different wind conditions, but also due to different incidence angles and noise floors. In the following, we look closer into the feature displaying the strongest correlation to $f_{MP}$ in each of the scenes, $R_{VV/HH}$ in T3 and $\sigma^0_{VV}$ in T4.

### 4.2 Intermediate-wind case

In the intermediate-wind case of scene T3, $R_{VV/HH}$ was found to be the SAR feature with the strongest correlation to $f_{MP}$. Combining $f_{MP}$ retrieved from the 43 helicopter images covering the investigated floe with $R_{VV/HH}$ extracted from the corresponding areas in scene T3, we see an increase in $R_{VV/HH}$ with $f_{MP}$ in Fig. 4, as well as a large variability between the samples. Grey dots correspond to areas with some degree of sea ice deformation, while blue dots correspond to areas with completely level ice. Deformation information is extracted from visual inspection of the helicopter images. The partly negative values of $R_{VV/HH}$ imply that $\sigma^0_{HH} > \sigma^0_{VV}$. This might be a result of multiple scattering events in the sea ice volume or sea ice surface, possibly connected with sea ice deformation. A majority of the lowest $R_{VV/HH}$ values are appearing in partly deformed areas. Areas with some degree of deformation also represent the lowest $f_{MP}$. A robust least squares linear fit is applied to the scatter plot, displaying a relationship of:

$$f_{MP}(R_{VV/HH}) = 0.49 \cdot R_{VV/HH}(dB) + 0.30. \tag{17}$$

The goodness of fit of the regression is reflecting large sample variation, with $R^2_{fit} = 0.21$ and $RMSE = 0.40$. This implies a weak correlation, corresponding well to the Spearman's correlation of 0.45 (Table 4). However, the co-location between the helicopter images and the sea ice floe contain some uncertainty (a maximum areal offset of $27\%$) possibly introducing a random error to the regression, resulting in an artificially low $R^2_{fit}$.

Applying the regression fit from on Eq. 17 to the full floe in scene T3 results in the regression fit probability density distributions (PDFs) presented in the top panel of Fig. 5. The results are presented both for a $21 \times 21$ and a $51 \times 51$ pixels smoothing window, corresponding to areas of $50 \times 40$ m and $120 \times 95$ m in the across $\times$ along flight direction. Observed distributions of $f_{MP}$ retrieved from the

43 images covering the floe (floe) and from images in all included flights (region), are also included in the figure. Statistics of the distributions are given in Tables 2 and 5. The regional distribution has a slightly higher mean than the floe distribution. Due to the few samples of the floe distribution, we consider the regional distribution more appropriate for comparison with the regression fit distributions. Employing the regression fit with a $21 \times 21$ pixels smoothing window, equaling the areal size of the helicopter images, results in a mean close to the observed regional distribution. The regression distribution is however too wide compared to the observed ones, reflecting the large sample variation seen in Fig. 4. Speckle (noise like interference between scatterers within a resolution cell) in the SAR image might explain the wider distribution. Increasing the smoothing window size reduces speckle, and a better correspondence between the width of the regression and observed distributions is achieved by employing a $51 \times 51$ pixels window. The bottom panel of Fig. 5 displays $f_{MP}$ estimated for the floe in T3 based on eq.17 with a $51 \times 51$ pixels window. Open water is masked out. The estimation shows a highly spatially variable $f_{MP}$, with few homogenous areas. Areas of deformed sea ice displayed with bright colors in Fig. 2 cannot be recognized, even if these areas are expected to have a lower $f_{MP}$.

Zooming in to the southern part of the area covered by the helicopter survey on the floe in T3, Fig. 6 displays $f_{MP}$ estimated from Eq. 17 with the observed $f_{MP}$ from the helicopter images overlaid. Two different pixels smoothing windows are shown ($21 \times 21$ and $51 \times 51$). Note that the center pixel underlying each helicopter image frame would give the most representative value for comparison to the observed $f_{MP}$, as pixels closer to the frame contain a larger amount of information from outside the frame. The middle panel displays the mean estimated $f_{MP}$ value for each frame together with the observed $f_{MP}$ values along the track. The maps confirm some overlap between the estimated and observed $f_{MP}$, but also illustrates that there is room for improvement. The estimation with a $51 \times 51$ pixel smoothing window appears less variegated than the $21 \times 21$ estimation, and the range of the estimated $f_{MP}$ values also corresponds better to those observed from the helicopter images in the $51 \times 51$ estimation.

Applying the regression fit from Eq. 17 with a $51 \times 51$ pixel window to the four full SAR scenes included in our study reveals a high correlation between the regression fit distribution and the observed regional $f_{MP}$ distribution for T3 (see Fig. 7 and Tables 2 and 5). On the full scene scale, the regression fit manages to reproduce both the mean and the standard deviation of the regional distribution representative for the area. Scene T1 and T2 are acquired at $\sim 8°$ higher incidence angle than scene T3, and $f_{MP}$ is slightly overestimated in these scenes. From Fig. 7, the overestimation is lower for scene T1 than for T2, possibly reflecting the low wind speed at acquisition of T1 (Tab 1). The least consistency between the regression fit distribution and the observed distribution is, as expected, found for scene T4, confirming the results shown in Table 4 and Fig. 3.

### 4.3 Low-wind case

In the low-wind case of scene T4, $\sigma^0_{VV}$ was found to have the strongest correlation to $f_{MP}$ among the investigated SAR features. Combining $f_{MP}$ retrieved from the 43 helicopter images covering the floe with $\sigma^0_{VV}$ extracted from the corresponding areas in T4, we see a decrease in $\sigma^0_{VV}$ with $f_{MP}$ in Fig. 8. A large variability between the samples can be observed. Grey dots correspond to partly deformed areas, while blue dots represent level ice. As for the intermediate wind case, a robust least square linear fit was applied to the data to describe the relationship between $\sigma^0_{VV}$ and $f_{MP}$:

$$f_{MP}(\sigma^0_{VV}) = -52.83 \cdot \sigma^0_{VV} + 1.89. \tag{18}$$

Note that $\sigma^0_{VV}$ is not in dB. Again, the goodness of fit of the regression is reflecting large sample variation, with $R^2_{fit} = 0.26$ and $RMSE = 0.0039$.

Estimated $f_{MP}$ PDFs based on Eq. 18 for the full floe in scene T4 are presented in the top panel of Fig. 9 together with observed distributions from the floe and from all flights included in the study. The results are presented both for a $21 \times 21$ and a $51 \times 51$ pixels smoothing window, corresponding to areas of $65 \times 30$ m and $155 \times 65$ m in the across $\times$ along flight direction. The regression fit distributions give a good reproduction of the observed mean (see Tables 2 and 5). As in the intermediate-wind case, a smoothing window of $51 \times 51$ pixels results in a distribution width closer to the observed than a $21 \times 21$ pixels window. The $\sigma^0_{VV}$-based estimation of $f_{MP}$ with a $51 \times 51$ smoothing window for the full floe in scene T4 result in a large spatial variability in $f_{MP}$ (see bottom panel of Fig. 9). In contrast to the $f_{MP}$ estimation based on $R_{VV/HH}$ for the floe in scene T3 (Fig. 5), the estimation based on $\sigma^0_{VV}$ partly manages to produce lower $f_{MP}$ in areas with deformed sea ice.

Figure 10 shows $f_{MP}$ estimated from Eq. 18 with the observed $f_{MP}$ from the helicopter images overlaid for two different pixels smoothing windows ($21 \times 21$ and $51 \times 51$). Note that the center pixel underlying each helicopter image frame would give the most representative value for comparison to the observed $f_{MP}$. To illustrate this, the middle panel shows the mean estimated $f_{MP}$ value for each frame together with the observed $f_{MP}$ values along the track. In general, a good overlap between the estimated and observed $f_{MP}$ can be seen, even though some scatter exists. As in Fig. 6, the estimation with a $51 \times 51$ pixel smoothing window appears less variegated than the $21 \times 21$ estimation, and the range of the estimated $f_{MP}$ values also corresponds better to those observed from the helicopter images in the $51 \times 51$ estimation than to those in the 21 x 21 estimation.

Investigating the regression fit's capacity of estimating $f_{MP}$ in the 4 full satellite scenes included in the study reveals that it is only applicable to give a good estimate in scene T4 (see Fig. 11 and Table 2 and 5)). In the three other scenes the estimate is poor; it underestimates $f_{MP}$, introducing negative fractions. Incidence angle correction according to Eq. 16 is applied to the figure, accounting for $\sigma^0_{VV}$ decrease with incidence angle.

## 5 Discussion

The results of this study show that $f_{MP}$ influences the signature of several X-band polarimetric features. The strongest correlations were found for $R_{VV/HH}$ and $\sigma^0_{VV}$, where linear regression fits gave $R^2_{fit}$ values of 0.21 and 0.26, respectively. These correlations are not strong enough for the results to be used directly in operational models. However, with improved methods and more satellite data added, our results imply a future potential in retrieving $f_{MP}$ from X-band SAR. For comparison, the method developed for retrieval of $f_{MP}$ from MODIS has $R^2_{fit}$ values ranging from 0.28 to 0.45 (Rösel et al., 2012). As in C-band, parameters like wind speed, incidence angle, surface roughness, and SAR scale and resolution will affect the interpretation of the polarimetric melt pond signature of a X-band SAR scene. In the following, these factors will be discussed based on the results.

Accurate information about wind speed at the time of scene acquisition is crucial in $f_{MP}$ retrieval from SAR. In scene T3, the intermediate wind speed at acquisition ($U_2 = 6.2$ m/s) allowed for backscatter from the melt ponds, making use of $R_{VV/HH}$ for $f_{MP}$ estimation possible. In X-band, the Bragg criterion is exceeded for $s_{RMS} > 1.4$ mm. Scharien et al. (2014b) finds that melt ponds exceed this roughness at wind speeds above $U_{10} = \sim 5$ m/s, reducing the expected correlation between $R_{VV/HH}$ and $f_{MP}$ above this wind speed. This indicates that even better results could be achieved at lower wind speeds, but it also leaves a very narrow wind speed interval for melt pond retrieval with X-band SAR. Scene T4 represents a low wind speed situation ($U_2 = 0.6$ m/s), and our results indicate specular, or close to specular reflection from the melt ponds in this case. The weak melt pond backscatter, combined with a low SNR, hamper the use of difference in polarimetric properties between sea ice and melt ponds for melt pond fraction retrieval. The weak correlation seen between $R_{VV/HH}$ and $f_{MP}$ in Table 4 is most probably reflecting slightly different sea ice surface types surrounding the ponds in areas with low and high melt pond fraction, rather than different polarimetric signatures between melt ponds and sea ice. The low correlation observed between $R_{VV/HH}$ and $f_{MP}$ in this low wind case is in agreement with findings in Scharien et al. (2012, 2014b), while Scharien et al. (2014a) found $R_{VV/HH}$ to increase with $f_{MP}$ even at low wind speeds ($U_10 = 1.1$ m/s). Different wind speeds, incidence angle and sea ice types could all contribute to the deviating findings. The lack of backscatter from the melt pond surfaces compared to the sea ice could potentially be used for $f_{MP}$ retrieval utilizing $\sigma^0$, as the backscatter intensity becomes weaker with increasing $f_{MP}$. This is confirmed by Han et al. (2016), suggesting $\sigma^0$ to be a key feature in $f_{MP}$ estimation for MYI in X-band during calm winds. On the other hand, our results deviate from findings in C-band, where no correlation was found between $\sigma^0_{HH}$ and $f_{MP}$ at low wind speeds ($U_10 = 1.5$ m/s) by Yackel and Barber (2000).

Medium to high incidence angles ($> 35°$) have been found most suitable for $R_{VV/HH}$-based retrieval of $f_{MP}$ in C-band (Scharien et al., 2012, 2014b). In our study we found a significant correlation between $R_{VV/HH}$ and $f_{MP}$ at an incidence angle of $29°$ (T3), demonstrating that $f_{MP}$ has an impact on polarimetric X-band SAR signatures also at lower incidence angles. Scene T1 and T2 are

acquired at higher incidence angles ($36.9°$ and $37.9°$) than T3. In these two scenes, $f_{MP}$ is overesti-
mated by the $R_{VV/HH}$-based regression fit developed for scene T3. This is consistent with Scharien
et al. (2014b), showing an increase in $R_{VV/HH}$ with increasing incidence angle for melt ponds in
C-band. In the same study, $R_{VV/HH}$ for bare ice was not found to increase with incidence angle. The
difference in estimated $f_{MP}$ between scene T1 and T2 is most likely related to the low wind speed

in T1, which is below the expected wind speed limit for $f_{MP}$ estimation based on $R_{VV/HH}$ in both
C and X-band (Scharien et al., 2012, 2014b). However, the different acquisition geometry observed
in Fig. 1 could also play a role. At an incidence angle of $44°$, a considerable part of the backscatter
signal was below the noise floor in our study. The low signal-to-noise ratio of TerraSAR-X limits
$f_{MP}$ retrieval based on $R_{VV/HH}$ at high incidence angles, leaving the suitable range of incidence

angles smaller than for Radarsat -2 (Scharien et al., 2014a). The accuracy of $f_{MP}$ estimation based
on $\sigma_{VV}^0$ is also strongly dependent on incidence angle, as $\sigma_{VV}^0$ in general decreases with increasing
incidence angle for sea ice. The underestimation of $f_{MP}$ in scenes T1-T3 is likely related to higher
wind speeds at the time of acquisition.

    The Brag criterion ($ks < 0.3$) is exceeded when $s_{RMS} > 1.4$ mm in X-band. The surface rough-

ness estimations performed during the ICE2012 campaign indicates that the sea ice in the study
region exceeds this criterion, introducing a roughness dependency of $R_{VV/HH}$. This is in agreement
with previous findings in the study region (Beckers et al., 2015), but deviates from findings reported
by Scharien et al. (2014b), where fast ice at the Central Canadian Archipelago partly filled the crite-
rion in X-band. From the helicopter images, some of the very low $R_{VV/HH}$ values observed at the

investigated floe in scene T3 were from slightly deformed areas, possibly explaining the negative
ratios. However, no general trend in low $R_{VV/HH}$ values in deformed areas was found in our study.
Other small scale surface scattering processes could also have caused the low $R_{VV/HH}$, negative
values have also been reported in other FYI studies, e.g., Geldsetzer and Yackel (2009)and Scharien
et al. (2014b). Multiple scattering events in the sea ice surface and sea ice volume may also have

contributed to the large sample variations observed in Figs. 4 and 8. Detailed surface roughness mea-
surements combined with $f_{MP}$ observations are needed to further investigate the influence of sea
ice surface roughness on $f_{MP}$ based on $R_{VV/HH}$.

    The smoothing window size used for direct comparison between $f_{MP}$ retrieved from the heli-
copter images and the polarimetric SAR features was appointed by the areal coverage of the heli-

copter images in our study. However, a $40 \times 60$ m window (corresponding to $21 \times 21$ pixels) might
not be the ideal scale of investigation. Advancing the regression fits suggested in our study to the full
floe or full scenes with a larger window ($51 \times 51$ pixels) gave better reproductions of the width of
the $f_{MP}$ distribution retrieved from the helicopter images. A larger window size reduces the amount
of speckle in the SAR scenes, which possibly explains the improvement. Even larger window sizes

were used in Scharien et al. (2014a), estimating $f_{MP}$ based on $R_{VV/HH}$ in a $7.5 \times 7.5$ km grid
from C-band Radarsat-2. Opposite to this, Han et al. (2016) found a $15 \times 15$ pixels window to give

the best estimate of mean $f_{MP}$ based on a combination of several SAR features in a TerraSAR-X scene. In climate applications, $f_{MP}$ estimation from a full scene is more applicable than estimation from small areas within the scene. The large sample variability observed in Fig. 4 might therefore
be negligible, as long as the $R_{VV/HH}$-based regression fit produces a good estimate of the mean $f_{MP}$ for a larger area. A wider study of the influence of scale on SAR $f_{MP}$ retrieval is needed in the future.

    In addition to $R_{VV/HH}$, five other dual-polarimetric SAR features were included in our study, most of these showed no statistical significant relationship to $f_{MP}$ in our data set. This is also an
important result, implying useful knowledge for instance in classification of summer sea ice based on X-band imagery. The statistical feature $RK$ showed a promising relation to $f_{MP}$ in C-band on fast ice in the Fram Strait (Fors et al., 2015), but no relation was found in our investigation. Lack of the HV-channel, or less dominant height difference between ponds and sea ice could both possibly explain the absence of correlation. $\alpha'_1$ was found significantly correlated with $f_{MP}$ in scene T3.
This is likely a result of the expected relation between $\alpha'_1$ and $R_{VV/HH}$ (van Zyl and Kim, 2011). In scene T4, several of the polarimetric SAR features were found related to melt pond fraction before NESZ subtraction, after NESZ subtraction, only the standard deviation of $|\rho|$ showed a relationship. This indicates that the correlations only reflected the low signal-to-noise ratio of the scene, as has previously been described in oil/water discrimination (Minchew et al., 2012).

The findings in our study deviate from the findings of Han et al. (2016), where $\sigma^0_{HH}$, $\angle \rho$ and $\alpha'_1$ were found to be the most prominent polarimetric features in separating melt ponds, sea ice and open water in high resolution X-band SAR imagery. Differences in sea ice type, sea ice surface roughness, wind conditions and SAR incidence angle could possibly explain why different polarimetric features are sensitive to $f_{MP}$ in the two studies. The methods of the two studies are also slightly different, as
Han et al. (2016) classify each pixel into melt pond, sea ice or open water, while our study focuses on mixtures of melt ponds and sea ice. Exact wind information lacks in Han et al. (2016), but the wind speed is expected to be low. This could explain why $\sigma^0_{HH}$ contributes strongly in $f_{MP}$ estimation, and is then in accordance to our findings. The diverging results in the two studies emphasize the need of investigating melt ponds impact on SAR imagery under different conditions and for a variety of
sea ice types. It also stresses the importance of supplementary measurements of parameters like wind speed and sea ice surface roughness.

    The correlations found in our study are not very strong. The weak to moderate correlations might suggest a limited sensitivity to $f_{MP}$ in X-band SAR imagery, but they could also reflect limitations in the data set. The co-location between the helicopter images and the SAR imagery is estimated to have
a possible offset of at most 27%, potentially introducing a large random error into our investigation, lowering the correlation values. A larger degree of smoothing than the area covered by the helicopter images allows for might also be needed to improve the results. The absolute radiometric accuracy of TSX scenes could also influence the results of our study, but this influence is expected to be very

small compared to other uncertainties. All the above-mentioned issues should be addressed in future studies.

## 6 Conclusions

Melt ponds play an important role in the sea-ice-ocean energy budget, but the evolution of melt pond fraction ($f_{MP}$) through the melt season is poorly monitored. Satellite-borne polarimetric SAR has shown promising results for $f_{MP}$ retrieval in C-band, but few studies have investigated the opportunities in X-band. In this study we demonstrate statistically significant relations between $f_{MP}$ and several polarimetric SAR features on drifting FYI in X-band, based on helicopter-borne images of the sea ice surface combined with four dual polarimetric SAR scenes. The study reveals a prospective potential for $f_{MP}$ estimation from X-band SAR, but also stresses the importance of including wind speed and incidence angle in a future robust $f_{MP}$ retrieval algorithm. Such an algorithm could supplement optical methods, and be used as a tool in climate applications, e.g, in studies of melt pond evolution mechanisms.

$R_{VV/HH}$ was found to be the most promising SAR feature for $f_{MP}$ estimation in our study, in agreement with previous findings in C-band. Hence, both the HH and VV polarimetric channels are needed for a future $f_{MP}$ retrieval in X-band. The theoretical range of suitable wind speeds ($< 5$ m/s) and sea ice surface roughnesses ($s_{RMS} < 1.4$ mm) for $f_{MP}$ extraction based on $R_{VV/HH}$ are slightly more limited in X-band than in C-band, but our results show that $f_{MP}$ also influences the X-band SAR signature when these criteria are partly exceeded. The high noise floor of TerraSAR-X also restricted use of scenes with incidence angles above $\sim 40°$, while an incidence angle of $29°$ gave better results. Future studies should focus on incidence angles in the range between $29°$ and $40°$. At very low wind speeds (0.6 m/s), the backscatter signal from the melt ponds became too low for $f_{MP}$ retrieval based on polarimetric features due to specular reflection. In that case, $\sigma_{VV}^0$ was found suitable for $f_{MP}$ estimation due to the lower backscatter intensity in areas with high $f_{MP}$. In the future, use of X-band scenes can possibly increase the total amount of SAR data accessible for $f_{MP}$ retrieval, despite their limitations compared to C-band scenes.

An extended amount of *in situ* and airborne measurements together with satellite scenes are needed to establish robust $f_{MP}$ estimation algorithms for X-band SAR. Information about wind speed is crucial for $f_{MP}$ retrieval, and can be retrieved from existing meteorological models or autonomous buoys measuring wind speed, where no ship or camp is present. Challenges in co-location of airborne observations and SAR imagery limited coordinated use of existing data in our study and introduced uncertainties in our results, with areal offsets of up to $27\%$, possibly causing artificially low correlation values. Better co-location, for instance through corner reflectors or GPS senders located in the specific study area, should be aimed for in future studies. With a shift towards more

seasonal drifting FYI, it is important to include this sea ice type in the studies, despite difficulties in comparing *in situ* and airborne measurements with satellite SAR scenes during drift.

Our study only investigates a few SAR scenes under similar sea ice conditions, and the ability of the suggested regression fits to predict changes in $f_{MP}$ is not included. This is an important aspect. For development of a robust operational method, future studies should aim to include a larger number of satellite scenes acquired with various sea ice conditions, melt pond evolution stages, wind speeds, and incidence angles. The effect and limitations of sea ice surface roughness and dependency on

filtering size and scale should also be further investigated.

*Acknowledgements.* The authors would like to thank the captain, crew and scientists from the Norwegian Polar Institute (NPI) and Airlift AS on-board R/V Lance during the expedition ICE2012 for support and data collection. The TerraSAR-X data are provided by InfoTerra. We acknowledge S. Hudson at NPI for help with meteorological data, and A. Fransson, also at NPI, for providing ice core information. Thanks to W. Dierking

at the Alfred Wegner Institute and C. Brekke and T. Eltoft at Department of Physics and Technology, UiT-The Arctic University of Norway for participation in discussions, and to S. N. Anfinsen at Department of Physics and Technology, UiT-The Arctic University of Norway for useful comments on the manuscript. The project was supported financially by Regional Differensiert Arbeidsgiveravgift (RDA) Troms County, by the project "Sea Ice in the Arctic Ocean, Technology and Systems of Agreements" ("Arctic Ocean", subproject "CASPER")

of the Fram Centre, and by the Centre for Ice, Climate and Ecosystems at the NPI. The airborne data collection was also supported by ACCESS, a European Project within the Ocean of Tomorrow call of the European Commission Seventh Framework Programme, grant 265863.

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

**Table 1.** Overview of the satellite scenes.

| Date | Time (UTC) | Scene ID | Incidence angle | Pixel spacing (az. × ground range) | Wind speed (2 m.a.s.) | Air temperature (2 m.a.s.) |
|---|---|---|---|---|---|---|
| 28 Jul 2012 | 06:52 | T1 | $36.9°$ | 2.4 m × 1.5 m | 1.6 m/s | $0.1°C$ |
| 29 Jul 2012 | 14:25 | T2 | $37.9°$ | 2.5 m × 1.5 m | 5.1 m/s | $1.1°C$ |
| 31 Jul 2012 | 13:51 | T3 | $29.4°$ | 2.4 m × 1.9 m | 6.2 m/s | $-0.8°C$ |
| 2 Aug 2012 | 14:51 | T4 | $44.2°$ | 3.0 m × 1.3 m | 0.6 m/s | $0.8°C$ |

**Table 2.** Overview of the images captured during the helicopter flights. Only images without open water fraction are included in the study. The bottom entries show the regional values derived from all five flights, and the local values of the floe investigated in T3 and T4.

| Date | Time (UTC) | No. of images | Transect length | Mean $f_{MP}$ | Std. $f_{MP}$ |
|---|---|---|---|---|---|
| 31 Jul 2012 | 7:36-8:10 | 848 | 67 km | 30.1% | 10.0% |
| 1 Aug 2012 | 7:22-8:34 | 1364 | 139 km | 31.1% | 12.3% |
| 1 Aug 2012 | 16:45-18:03 | 1383 | 154 km | 34.8% | 12.8% |
| 2 Aug 2012 | 11:21-12:00 | 676 | 78 km | 33.0% | 13.7% |
| 2 Aug 2012 | 14:43-16:04 | 1458 | 170 km | 33.2% | 11.4% |
| Regional values | - | 5729 | 608 km | 33.2% | 11.4% |
| Floe values | - | 43 | 4 km | 30.6% | 11.1% |

**Table 3.** Estimated sea ice surface roughness ($s_{RMS}$) from five segments at the floe by R/V Lance. Values in parenthesis displays standard deviations (std) of $s_{RMS}$.

| Segment Nr. | Area | $s_{RMS}$ $(std(s_{RMS}))$ |
|---|---|---|
| 1 | 11000 m$^2$ | 6.7 (0.3) cm |
| 2 | 13530 m$^2$ | 11.0 (10) cm |
| 3 | 11670 m$^2$ | 7.4 (0.6) cm |
| 4 | 13820 m$^2$ | 9.0 (0.4) cm |
| 5 | 12380 m$^2$ | 10.0 (0.4) cm |

**Table 4.** Spearman's correlation coefficient ($r$) between $f_{MP}$ retrieved from the helicopter images at the investigated floe, and mean and standard deviation of the polarimetric SAR features from the corresponding area in T3 and T4. Bold indicate significant values within a $95\%$ confidence interval.

| SAR feature | r (T3) Mean | r (T3) Std. | r (T4) Mean | r (T4) Std. |
|---|---|---|---|---|
| $\sigma^0_{HH}$ | 0.04 | 0.10 | **-0.33** | -0.27 |
| $\sigma^0_{VV}$ | 0.21 | 0.09 | **-0.54** | **-0.54** |
| $R_{VV/HH}$ | **0.45** | 0.03 | **-0.31** | **-0.48** |
| $H$ | 0.11 | 0.25 | 0.22 | -0.17 |
| $\alpha_1$ | **0.40** | 0.00 | -0.24 | 0.11 |
| $RK$ | 0.07 | 0.07 | -0.15 | 0.08 |
| $|\rho|$ | -0.13 | 0.04 | -0.17 | **-0.44** |
| $\angle\rho$ | -0.14 | 0.10 | -0.08 | 0.12 |

**Table 5.** Statistics of modeled $f_{MP}$ distributions.

| Area | Window size (pixels) | $f_{MP}(R_{VV/HH})$ Mean | $f_{MP}(R_{VV/HH})$ Std. | $f_{MP}(\sigma^0_{VV})$ Mean | $f_{MP}(\sigma^0_{VV})$ Std. |
|---|---|---|---|---|---|
| T3, floe | $21 \times 21$ | 34.9% | 24.8% | - | - |
| T3, floe | $51 \times 51$ | 35.0% | 11.0% | - | - |
| T4, floe | $21 \times 21$ | - | - | 30.6% | 26.0% |
| T4, floe | $51 \times 51$ | - | - | 31.4% | 16.7% |
| T1, full scene | $51 \times 51$ | 36.5% | 12.3% | 19.0% | 29.9% |
| T2, full scene | $51 \times 51$ | 45.1% | 13.3% | $-1.6\%$ | 27.8% |
| T3, full scene | $51 \times 51$ | 31.2% | 11.2% | 19.7% | 29.7% |
| T4, full scene | $51 \times 51$ | 51.9% | 12.3% | 36.3% | 15.7% |

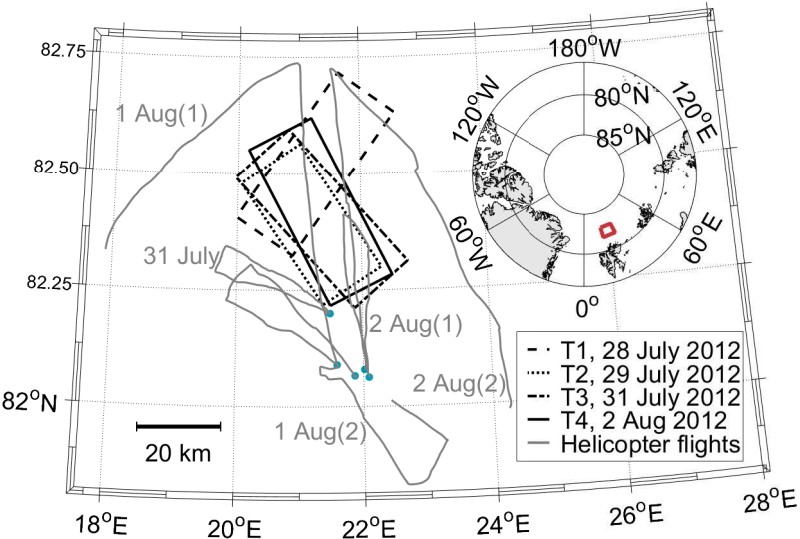

**Figure 1.** Map of the study area north of Svalbard, showing the location of the satellite scenes and the track of the helicopter flights. Blue dots mark the starting points of the flights. The red box in the inset map of the northern hemisphere shows the geographical position of the area displayed.

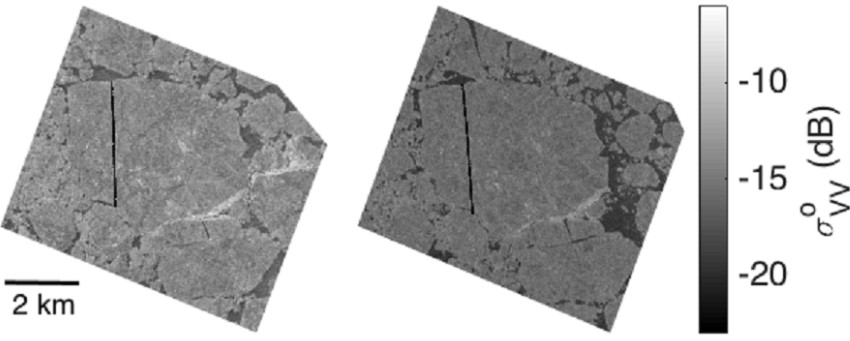

**Figure 2.** The floe investigated in scene T3 (left) and T4 (right) with a $11 \times 11$ pixels smoothing window. The black line marks the transect along which the helicopter image were taken.

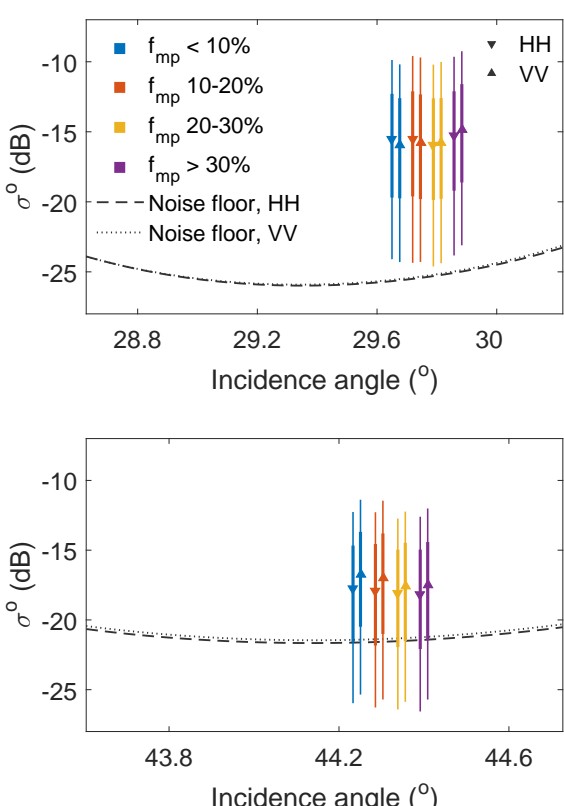

**Figure 3.** Signal-to-noise analysis of HH and VV channels for areas with different $f_{mp}$ retrieved from the investigated floe in scene T3 (top) and T4 (bottom). The triangles displays the median of $\sigma^0_{HH}$ (dB) (upward pointing) and $\sigma^0_{VV}$ (downward pointing). The thin line represents the part of $\sigma^0$ falling between the 10 and the 90 percentile, while the thick line represents the part of $\sigma^0$ falling between the 25 and 75 percentile. Hence, the lines indicate the distributions. All markers are offset from the middle position for clarity.

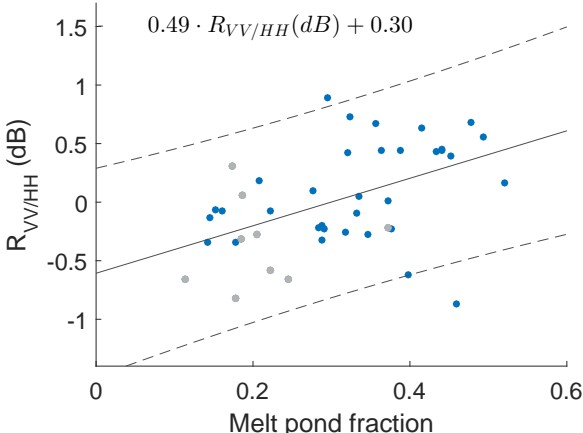

**Figure 4.** Scatter plot displaying $f_{MP}$ retrieved from the 43 helicopter images covering the investigated floe in T3, and mean $R_{VV/HH}$ extracted from the corresponding areas. Grey dots represent areas with partly deformed sea ice, wile blue dots represent areas of level ice. The trend line represents a robust bisquare weights least squares linear fit of the data, and the dotted line represent the 95% confidence interval of the regression. $R_{fit}^2$ equals 0.21.

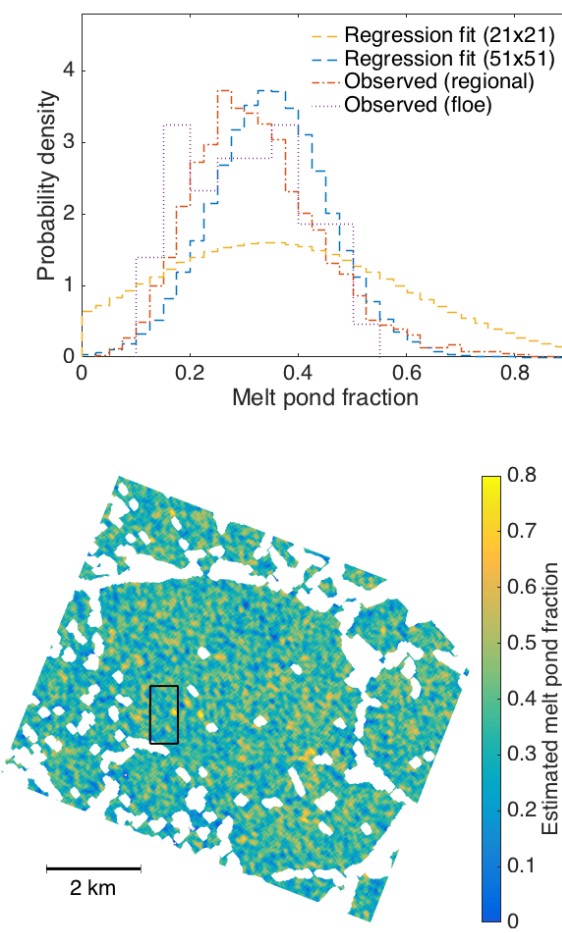

**Figure 5.** Top: Probability density distributions of $f_{MP}$ for the investigated floe in T3. Curves represent distributions produced by the regression fit based on $R_{VV/HH}$ with $21 \times 21$ and $51 \times 51$ pixels windows, and observed distributions from all helicopter flights (regional) and from the specific floe (floe). Bottom: Estimated $f_{MP}$ from the $R_{VV/HH}$ based regression with a $51 \times 51$ pixels window for investigated floe in T3. The frame outlines the area displayed in Fig. 6.

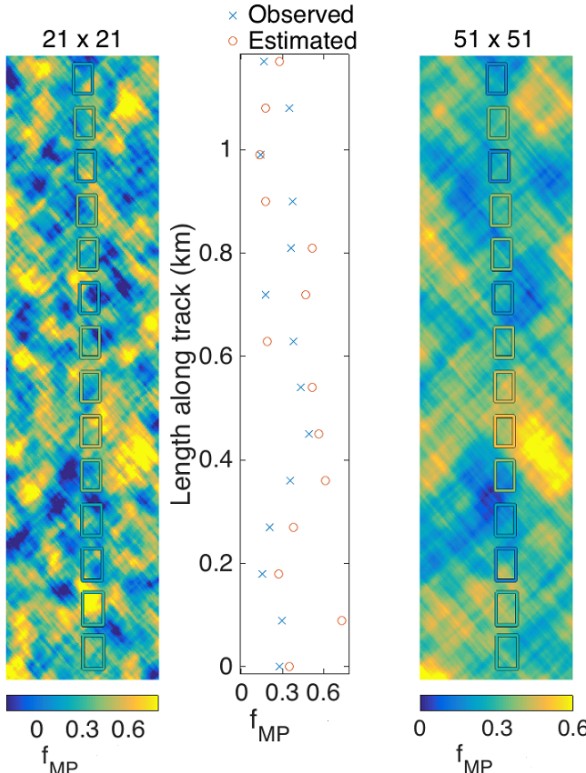

**Figure 6.** Melt pond fraction ($f_{MP}$) estimated from $R_{VV/HH}$, with the observed $f_{MP}$ from the helicopter images overlaid as colored frames. The area displayed is $0.3 \times 1.1$ km and its position is outlined with a frame in Fig. 5. The estimation is performed with $21 \times 21$ (left) and $51 \times 51$ (right) pixels windows. Note that the center pixel underlying each helicopter image frame would give the most representative value for comparison to the observed $f_{MP}$, as pixels closer to the frame contain a larger amount of information from outside the frame. The middle panel displays the mean estimated $f_{MP}$ value for each frame together with the observed value.

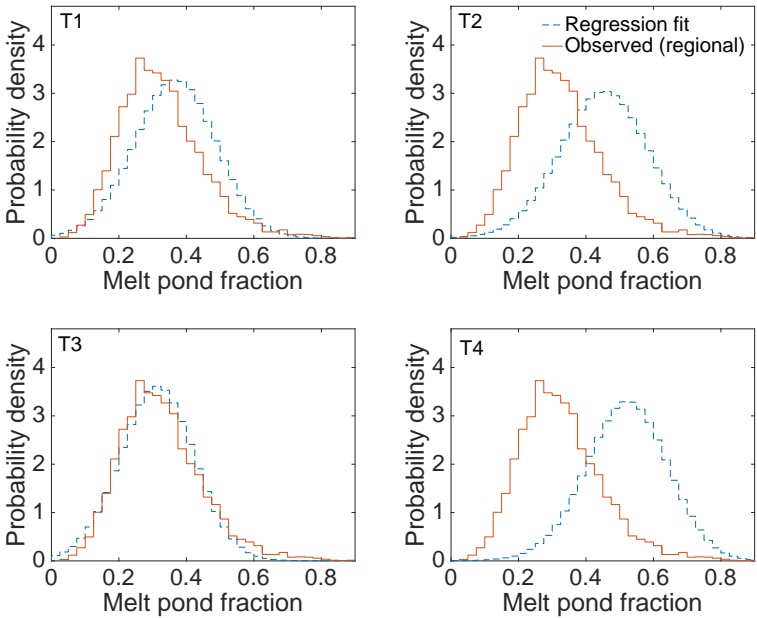

**Figure 7.** Probability density distributions of $f_{MP}$ for the four investigated scenes (T1-T4). Curves represent distributions produced from the $R_{VV/HH}$ based regression fit with a $51 \times 51$ pixels window, and the observed distribution retrieved from all five helicopter flights.

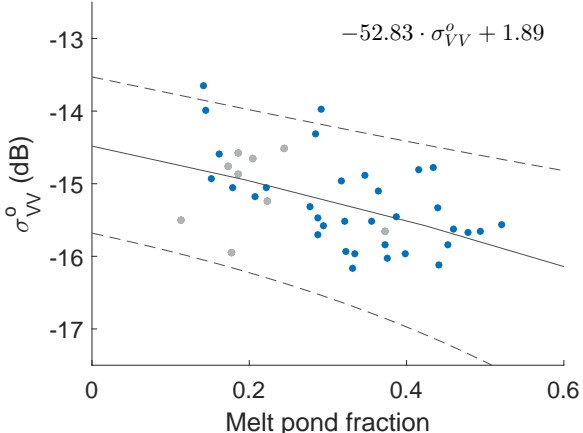

**Figure 8.** Scatter plot displaying $f_{MP}$ retrieved from the 43 helicopter images covering the investigated floe in T4, and mean $\sigma_{VV}^0$ extracted from the corresponding areas. Grey dots represent areas with partly deformed sea ice, while blue dots represent areas of level ice. The trend line represents a robust bisquare weights least squares linear fit of the data, and the dotted line represent the 95% confidence interval of the regression. $R_{fit}^2$ equals 0.26.

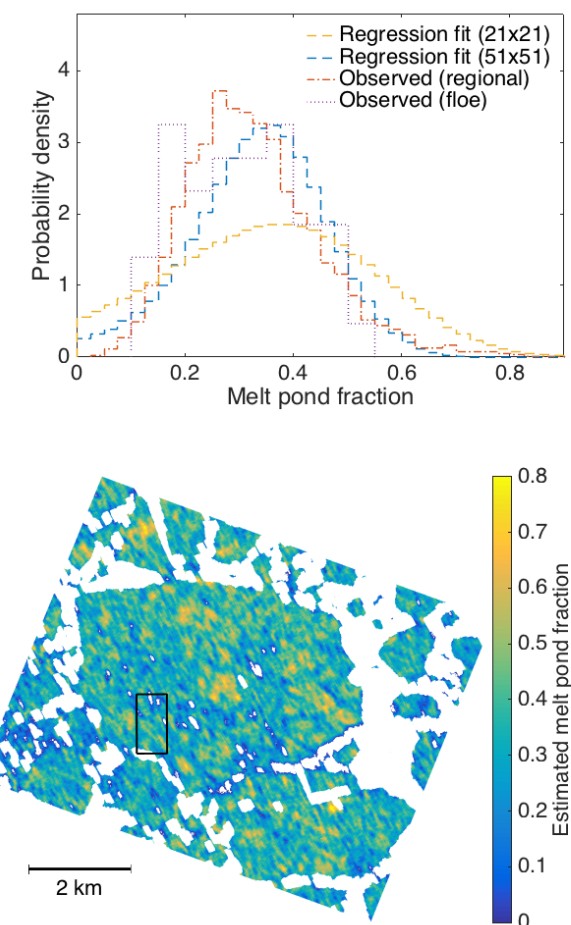

**Figure 9.** Top: Probability density distributions of $f_{MP}$ for the investigated floe in T4. Curves represent distributions produced by the regression fit based on $\sigma^0_{VV}$ with $21 \times 21$ and $51 \times 51$ pixels windows, and observed distributions from all helicopter flights (regional) and from the specific floe (floe). Bottom: Estimated $f_{MP}$ from the $\sigma^0_{VV}$ based regression with a $51 \times 51$ pixels window for investigated floe in T4.The frame outlines the area displayed in Fig. 10.

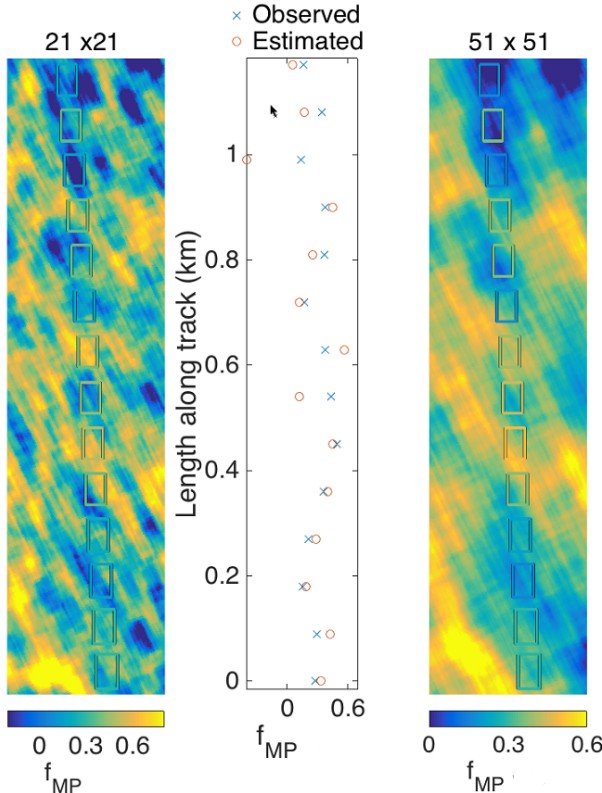

**Figure 10.** Melt pond fraction ($f_{MP}$) estimated from $\sigma^0_{VV}$, with the observed $f_{MP}$ from the helicopter images overlaid as colored frames. The area displayed is $0.3 \times 1.1$ km and its position is outlined with a frame in Fig. 9. The estimation is performed with $21 \times 21$ (left) and $51 \times 51$ (right) pixels windows. Note that the center pixel underlying each helicopter image frame would give the most representative value for comparison to the observed $f_{MP}$, as pixels closer to the frame contain a larger amount of information from outside the frame. The middle panel displays the mean estimated $f_{MP}$ value for each frame together with the observed value.

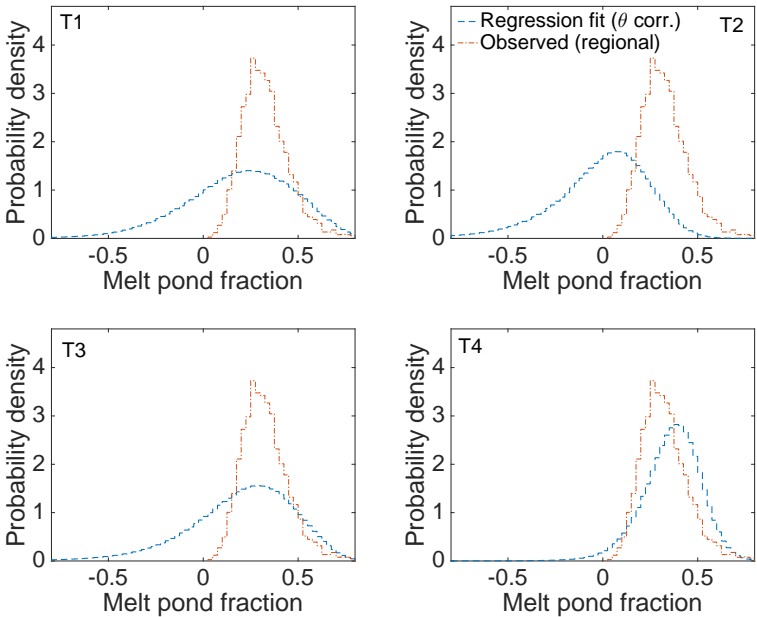

**Figure 11.** Probability density distributions of $f_{MP}$ for the four investigated scenes (T1-T4). Curves represent distributions produced from the $\sigma_{VV}^0$ based regression with a $51 \times 51$ pixels window, and the observed distribution retrieved from all five helicopter flights.