# Peer review of "Signature of Arctic first-year ice melt pond fraction in X-band SAR imagery"

_The Cryosphere, 2016_

## Referee Comment (RC1) · Anonymous Referee #1 · 3 Sep 2016

The manuscript is dedicated to the melt pond fraction estimation from X band of polarimetric SAR. This sensor is not dependent on cloud cover or presence of daylight, which is an advantage over radiometers like MODIS and MERIS. Currently, melt ponds are poorly represented in the climate models, and the melt pond research is therefore an important topic which fits well into the scope of the journal. The paper is well written and the text extensively referenced. However, the manuscript still has some potential for improvement in the points listed below.

1) The study is limited to the drifting first year ice and to the X band only; in the Introduction, the authors give a very extensive literature review which reveals a massive amount of work already published regarding SAR X and C bands for melt ponds on many ice types for many locations, among which the landfast FYI and MYI. What is the motivation for this additional study for drifting ice and X band? Drifting FYI is a widespread

[Figure]

ice type indeed, but it features a variety of subclasses which calls for a robust method. What is the advantage of X band over other SAR bands for this challenging task?

2) The study is dedicated to the comparison of helicopter-borne imagery to the dual polarisation X band SAR data. Overall, 4 SAR scenes have been taken, to which the helicopter data were possibly accurately collocated. Nevertheless, the comparison data shows considerable scatter, the Spearman correlation was used instead of Pearson (could you please justify this), and the noise equivalent was subtracted. The authors are struggling to collect all the available signal which is over the noise floor and compare it to the airborne data. However, even with this cumbersome approach, the correlation coefficients of the developed empirical relationships are $R^2=0.15$ and 0.21, which is a very weak to weak correlation for Spearman. The authors state the surface deformation as a reason for the scatter and claim the correlation "significant" and enough to give a starting point to MPF evaluation, but the reviewer fails to see how it could work.

Under these circumstances, the quality of the developed method when applied to a variety of different X-SAR images of drifting ice (even with known wind speed) is very hard to estimate, even when the one smoothes out or grids the retrieved pond fractions to coarser resolutions.

3) The authors compare MPF distributions from airborne and retrieved from SAR data. To evaluate the quality of the results even better, it would be good to show also the spatial situation: the MPF retrieved from SAR plotted on a lat-lon map and the airborne reference data overplotted on the same map using same colorscale. Upon checking spatial features or spatial uniformity, the reader can make sure that the retrieved MPFs are not random numbers, but really correspond to the field situation.

The authors come to the conclusion that the dual polarimetric SAR data in X band can be used for melt pond estimate given the appropriate wind speed, incidence angle, surface deformation ranges and also upon extensive smoothing or even taking the

mean value over the whole scene.

The impact and importance of such a product is not sufficient for advancing our understanding on melt pond processes and can only serve as complementary data for other studies. Currently, the manuscript serves more as a fundamental study on the SAR features in X band and more displays limitations than advantages of the data.

I recommend to support the shown MPF results with possibly more SAR scenes and definitely show the spatial MPF maps to confirm the quality of the pond retrieval, or refocus the manuscript on signatures of various ice/pond types in X band without the actual MPF retrieval.

Technical:

- Please add the error bar of the empirical fit in Eq. (16) and (17) on the corresponding figures, this helps to estimate the quality of the MPF retrieval.

- please add the correlation coefficient values into the abstract and into figure captions where you present the empirical fits.

- I suggest to merge the subsection 4.1 Sea ice conditions into the subsection 3.1 Study region. Current section 4.1 logically fits better to 3.1.

---

## Referee Comment (RC2) · Anonymous Referee #2 · 5 Sep 2016

**Signature of Arctic first-year ice melt pond fraction in X-band SAR imagery**

Ane S. Fors, Dmitry V. Divine, Anthony P. Doulgeris, Angelika H. H. Renner, and Sebastian Gerland

**General comments**

The authors have studied melt pond fraction (MPF) estimation using TerraSAR-X dual-polarization SAR imagery acquired over drift ice north of Svalbard, and presented empirical models for MPF estimation in two different wind speed conditions (low speed and intermediate speed).

In the Introduction Section authors give good overview on importance of melt ponds on the Arctic sea ice heat budget and in the Arctic climate system, and on previous studies on melt pond fraction estimation with optical and SAR imagery. The number of previous SAR studies on melt pond detection and fraction estimation is quite large, and so far a generic method for the estimation has not been presented/developed. There has been some success for the melt pond fraction (MPF) estimation over smooth landfast ice using C-band co-polarization ratio (HH and VV pol SAR images needed) or HH-pol backscattering coefficient ($\sigma°$). For MPF estimation over drift ice only few studies has been conducted. At least it seems that MPF estimation over drift ice with C-band single pol imagery is not possible. Over drift ice sea ice deformation features like ice ridges and make MPF estimation in theory much more difficult than over smooth landfast ice. Other frequencies than C-band have been used only in few case studies. Likely (to my opinion) accurate MPF estimation is only possible with high resolution (<5-10 m) SAR imagery. So far time series of MPF maps over the Arctic have been produced only with optical imagery (MODIS, MERIS). These charts are limited by accuracy of automatic cloud masking and persistent cloud cover during the Arctic summer. Accurate MPF charts from SAR imagery would supplement greatly the MPF charts from optical imagery.

Section 2 discusses nicely about melt pond signatures in SAR imagery, but it could also include overview of observed $\sigma°$ behavior during melt ponding season (ponding, drainage etc.), see e.g. D. G. Barber, J. J. Yackel, and J. M. Hanesiak, "Sea ice, RADARSAT-1 and arctic climate processes: A review and update," Can. J. Remote Sens., vol. 27, no. 1, pp. 51–61, 2001.

Dual-polarization TerraSAR-X imagery acquired over drift ice for MPF estimation have been previously studied by Kim et al. (2013) and Han et al. (2016). Kim et al. (2013) used only one TSX image acquired in Aug 2011 over East Siberian Sea, but they have large amount of co-incident airborne very fine resolution X-band (single pol) images. Comparison MPF data was from airborne photography. Han et al. (2016) used the same datasets and also one additional TSX image acquired in July 2011 over the Chukchi Sea. Kim et al. (2013) estimated MPF with "We first delineate the ice and melt pond features using image processing software (ENVIA EX), based on the combination of multiscale segmentation and aggregation methods."; not discussed in more details, but Han et al. (2016) studied various polarimetric parameters and their textural features in MPF estimation by machine learning approaches.

The authors have used here four TSX dual-polarized StripMap images acquired during ICE2012 campaign in north of Svalbard. Comparison MPF data was from helicopter-borne optical imagery. In addition, surface roughness data was calculated from stereo camera imagery, and weather data was measured by R/V Lance at the ICE2012 campaign site. They have studied MPF estimation with different polarimetric parameters calculated from the dual-pol TSX imagery, as was done also by Han et al. (2016). The main questions now are: 1) Does this study give new scientific results/information compared to Kim et al. (2013) and Han et al. (2016)? 2) In what ways it is different to them, data and/or methods?

My answers: Study area is different (Chukchi Sea vs. Svalbard) which could have influence on the results if sea ice conditions (FYI vs. MYI) where different; authors should discuss this in the paper. Wind speed is taken into account here unlike in Kim et al. (2013) and Han et al. (2016). Wind speed has large effect on the backscattering from melt ponds (not frozen). Somewhat more SAR images have used, four here compared to two in Han et al. (2016), making the results here more reliable. The developed MPF algorithms are linear functions between MPF and one polarimetric parameter. I favor this kind of simple approach as the results can be related easily to theoretical backscattering models. Han et al. (2016) utilized machine learning approaches where relations between polarimetric parameters (and scattering theory) and an estimated parameter may not be very clear. However, I think that the paper in its current form gives quite little new information/findings compared to previous studies on the MPF estimation with SAR. The statistical reliability of the developed empirical MPF estimation models seems quite low, r2 was at best only 0.21 and RMSE is high. The value of the paper could be improved by following changes and additions:

The empirical models for the MPF estimation were developed using datasets over a large ice floe. Why were not all co-incident SAR imagery vs. airborne photography used? How results would change if they were?

Both wind speed and SAR incidence angle have large effect on the MPF estimation. Wind speed is now taken into account by MPF models for two different wind conditions. I suggest you developed MPF models which include incidence angle or compensate σ° incidence angle variation before MPF estimation. The study should include more variable wind speed conditions, but in the current dataset these are not present.

You could study effect of sea ice type in the MPF estimation, e.g. by first segmenting the SAR images to level ice and deformed ice categories (with the help aerial photography if possible). In best case we could have also sea ice type taken into account in the MPF estimation. You have also surface roughness data which could be utilized here.

Show MPF maps from some SAR images and discuss spatial variation present, does it make sense? You have four SAR images, how does estimated MPF behave temporally? Now Table 5 shows MPF averages over the full scenes, but these are not much discussed in the text, and temporal variation does not seem right (36.2->45.7->31.2->53.3).

Can you compare your estimates with those from optical imagery?
See http://icdc.zmaw.de/1/daten/cryosphere/arctic-meltponds.html

The study would benefit greatly from a larger SAR dataset. Are there any co-incident TSX vs. in-situ / airborne data from NICE2015 campaign you could use? You really need more wind speed conditions for the MPF estimation development. Even including more TSX images without corresponding comparison data is possible, you could study spatial and temporal trends. In addition, any fine resolution C-band images available? Comparison between C- and X-band would be nice addition.

In general, the paper is well written and structured, and easy to read and understand. The data processing and analysis methods are scientifically sound and discussed in needed detail. I am afraid in the current form the paper gives quite few new scientific findings compared to previous studies.

From Conclusions: "Future studies should aim to include a larger number of satellite scenes acquired during various sea ice conditions, melt pond evolution stages, wind speeds and incidence angles. The effect and limitations of sea ice surface roughness and dependency on filtering size and scale should also be further investigated."

You should consider taking some of these topics to this paper!

Finally, Yackel and Barber (2000) speculated that $\sigma°$ may be more closely related to the albedo than to melt pond fraction due to the fact that albedo results from the integration of all surface types (snow, saturated snow, melt ponds) which contribute to the measured $\sigma°$. What's the authors' view on this; would it be better to investigate the relationship between SAR data and albedo than SAR and melt pond fraction? Please, discuss this in Introduction Section.

**Specific comments**

**1. Introduction**

page 3, lines 90-92: terms 'dual polarimetric' and 'dual-polarisation' used, confusing…I think it should be 'dual-polarisation' for SAR imagery with two polarizations.

**2. Melt ponds in SAR imagery**

p. 4, l. 118: "Observed surface roughness increases with increasing frequency, making X-band more sensitive to small-scale surface roughness than C-band."

I think surface roughness is physical property of a surface, and its effect on backscattering depends on radar wavelength.

l. 124: "Six of these features are included in our study and are described in the following subsection."

In Table 4 there are eight features.

**3.2 Data set**

Give absolute calibration accuracy and equivalent number of looks (ENL) (i.e. effect of radar fading) in TSX images. Do they have any significant effect on your data analysis results?

**4.1 Sea ice conditions**

Was there any nighttime re-freezing on sea ice and melt pond surfaces which could have influenced backscattering signatures in T1 TSX image acquired at 06:52 UTC on 28 July?

**4.3 Intermediate-wind case**

p. 12, l. 358: "From visual inspection of the helicopter images, some of the lowest RVV/HH values origin from slightly deformed areas with a surface roughness possibly exceeding the Bragg criterion."

Please discuss how this sea ice condition leads to low RVV/HH.

---

## Author Comment (AC1) · 25 Nov 2016

The manuscript is dedicated to the melt pond fraction estimation from X band of po- larimetric SAR. This sensor is not dependent on cloud cover or presence of daylight, which is an advantage over radiometers like MODIS and MERIS. Currently, melt ponds are poorly represented in the climate models, and the melt pond research is therefore an important topic which fits well into the scope of the journal. The paper is well written and the text extensively referenced. However, the manuscript still has some potential for improvement in the points listed below.

1)The study is limited to the drifting first year ice and to the X band only; in the Introduction, the authors give a very extensive literature review which reveals a massive amount of work already published regarding SAR X and C bands for melt ponds on many ice types for many locations, among which the landfast FYI and MYI. What is the motivation for this additional study for drifting ice and X band? Drifting FYI is a widespread ice type indeed, but it features a variety of subclasses which calls for a robust method. What is the advantage of X band over other SAR bands for this challenging task?

As the reviewer highlights, many studies have already been exploring melt pond fraction retrieval from SAR. However, a majority of these studies are *single polarimetric* studies, mostly reaching vague results when it comes to melt pond fraction estimation from SAR. Only a few studies have investigated the *multi polarimetric* SAR signature of melt ponds. Among these, only one study (Han et. al 2016) has explored X-band opportunities. Han et al. (2016) focuses on one single satellite scene with known melt pond fractions, the study is performed on MYI, and very important factors like SAR incidence angle, noise floor and wind speed are not discussed. Hence, we believe there is need for more studies polarimetric X-band SAR signatures of melt ponds. To make this clearer in our manuscript, the following changes have been made:

- The following paragraph was added (Introduction, P3L90, P3L93): "In summary, the main achievements on $f_{MP}$ retrieval with SAR come from dual polarimetric C-band studies on land-fast FYI. The potential of $f_{MP}$ retrieval with polarimetric X-band SAR has only been explored in one single study by Han et al. (2016), focusing on MYI. Hence, there is a need for more studies on the influence of $f_{MP}$ on polarimetric X-band SAR imagery. As MYI and land-fast FYI have been the main focus in previous studies, there is also a need to expand to other sea ice types. Drifting FYI is becoming more prominent in the Arctic with the recent shift to a thinner, more seasonal, and more mobile sea ice cover (Perovich et al., 2015), and the polarimetric SAR signature of $f_{MP}$ in drifting FYI needs more attention."
- He following sentence was added (Introduction, P3L69, P3L71-72): "… underestimation of $f_{MP}$. All in all, retrieval of $f_{MP}$ from single polarimetric SAR has proven to be difficult."

TerraSAR-X offers higher resolution than Radarsat-2, which is expected to be an advantage, due to the small size of melt ponds. X-band is also offering sensitivity to smaller surface roughnesses than C-band, which will could potentially affect the polarimetric signature beneficially. In general, it is also

important to study the effect of melt pond fractions in all operational SAR frequencies, both because they can supplement each other, and because melt pond signatures will appear differently at different wavelengths. Knowledge of melt ponds polarimetric signatures is also important for classification of sea ice in X-band scenes. To highlight these points, the following changes have been introduced in the manuscript:

- The following sentence was added (Introduction, P3L93, P4L104-107): "TerraSAR-X offers very high resolution multi-polarimetric data, with a strong sensitivity to micro-scale surface roughness due to the high frequency. Both the high resolution and sensitivity to surface roughness can be advantages in $f_{MP}$ investigations."
- The following sentence was added (Discussion, P15L478, P18L580-581): "This is also an important result, implying useful knowledge for instance in classification of summer sea ice based on X-band imagery."

2)The study is dedicated to the comparison of helicopter-borne imagery to the dual polarisation X band SAR data. Overall, 4 SAR scenes have been taken, to which the helicopter data were possibly accurately collocated. Nevertheless, the comparison data shows considerable scatter, the Spearman correlation was used instead of Pearson (could you please justify this), and the noise equivalent was subtracted. The authors are struggling to collect all the available signal which is over the noise floor and compare it to the airborne data. However, even with this cumbersome approach, the correlation coefficients of the developed empirical relationships are $R^2=0.15$ and $0.21$, which is a very weak to weak correlation for Spearman. The authors state the surface deformation as a reason for the scatter and claim the correlation "significant" and enough to give a starting point to MPF evaluation, but the reviewer fails to see how it could work. Under these circumstances, the quality of the developed method when applied to a variety of different X-SAR images of drifting ice (even with known wind speed) is very hard to estimate, even when the one smoothes out or grids the retrieved pond fractions to coarser resolutions.

This point addresses several parts of the manuscript, and the reply is divided into 8 individual bulletpoints (a)-h)) found below.

a) Spearman's correlation coefficient was used instead of Pearson's correlation coefficient as it allows for non-linear relationships, broadening the range of detectable correlations. It is also less sensitive to outliers, which can be a problem in SAR scenes due to speckle. We have now clarified this in the manuscript:

- The following sentence was extended (Method, P9L287, P12L330-332): "A negative sign indicates an inverse relationship. Spearman's correlation coefficient assumes a monotonic relationship. It is used instead of Pearson's linear correlation coefficient, to allow for non-linear correlations. It is also less sensitive to outliers than Pearson's correlation coefficient."

We would like to note that Spearman's correlation in this study was used as a simple metric to identify SAR features potentially useful for further $f_{MP}$ reconstruction. In the latter procedure, however, an ordinary linear regression technique was used (see point d)). We consider that in the future studies when a sufficient information on the SAR $f_{MP}$ signature for a broad range of controlling factors have been accumulated, an application of more sophisticated multivariate techniques will be required to elaborate the model(s) robust enough for operational products. This is reflected in the manuscript:

- The following sentence was added (Discussion, P12L365, P16L508-511) "These correlations are not strong enough for the results to be used directly in operational models. However, with improved methods and more satellite data added, our results imply a future potential in retrieving $f_{MP}$ from X-band SAR."

b) "Significant" or "statistical significant" are used several times in the manuscript, referring to correlations that are significant within a 95% confidence interval. This is clearly stated in the Results-section in the version of the manuscript published in TCD (Results, P11L351, P12L375).

c) We see that working with the signal both with and without NESZ-subtraction can be confusing for the reader. We have therefor decided to use the NESZ-subtracted signal in the manuscript, and only include values without NESZ subtraction in parentheses in Table 4. This imply the following changes:

- The following sentences were rephrased (Method, P8L230, P9L270): "All scenes were converted to ground range and radiometrically calibrated to $\sigma^0$. The noise equivalent $\sigma^0$ (NESZ) was then subtracted."
- The following sentences were rephrased (Results, P11L326-334, P12L374-385): "Values significant within a 95% confidence interval are highlighted in bold, and values in parentheses show results before NESZ subtraction of the signal. In scene T3, $R_{VV/HH}$ shows the strongest correlation to $f_{MP}$. In addition, the mean of $\alpha_1$ is significantly correlated to $f_{MP}$. None of the other investigated SAR features are significantly correlated to $f_{MP}$ in scene T3. In scene T4, the mean values of $\sigma^0_{HH}$, $\sigma^0_{VV}$ and $R_{VV/HH}$ are significantly correlated to $f_{MP}$, the strongest correlation is found for, $\sigma^0_{VV}$. Some of the standard deviation values are also correlated to $f_{MP}$. Without NESZ subtraction in the calibration, however, almost all features are correlated to $f_{MP}$. The large difference before and after NESZ subtraction indicates that the signal is close to, or reaching the noise floor."
- The following sentence was rewritten (Discussion, P15L477, P18L579-581): In addition to $R_{VV/HH}$, five other dual-polarimetric SAR features were included in our study, after NESZ subtraction most of these showed no statistical significant relationship to $f_{MP}$ in our data set.
- The figure caption of Table 5 was updated: "Spearman's correlation coefficient ($r$) between $f_{MP}$ retrieved from the helicopter images at the investigated floe, and mean and standard deviation of the polarimetric SAR features from the corresponding area in T3 and T4. Bold indicate significant values within a 95 % confidence interval, and values in parentheses are retrieved before NESZ subtraction in the calibration process."
- Figures and Table 6 were updated, now presenting results after NESZ subtraction.

d) The correlation coefficients of R^2=0.15 and 0.21 represent the least square regression fits of Eqs. 16 and 17 and Figs. 4 and 7, and are presented in addition to Spearman's correlation coefficient (corresponding values of -0.53 an 0.45). To clarify this in the manuscript, the regression fit correlations are renamed $R_{fit}^2$ (changed in Results, P12L362, P13L398, P13L421, P15L477), while Spearman's correlation coefficient is kept as $r$. For clarification, the following change was introduced:

- The following sentence was changed (Abstract, P1L7, P1L7): Co-polarisation ratio was found to be the most promising SAR feature for melt pond fraction estimation at intermediate wind speeds (6.2 m/s), with a Spearman's correlation coefficient of 0.46."

e) Due to the reviewer's comments, we revised the regression fits, and managed to improve Eq. 17 by not log-transforming $\sigma^0_{VV}$, returning a new correlation coefficient of $R_{fit}^2$=0.26 (previously 0.15).

- Eq. 18 was changed in the text (Results, P13L397, P15L474): "$f_{MP}(\sigma^0_{VV})$=-52.83 $\sigma^0_{VV}$ +1.89"
- The following sentences were updated (Results, P13L398, P15L475): "Note $\sigma^0_{VV}$ is not in dB. Again, the goodness of fit of the regression is reflecting large sample variation, with $R_{fit}^2$=0.26 and RMSE=0.0039.
- Figures 7, 8 and 9 were updated according to the new regression fit equation.

f) We agree with the reviewer that these correlation values are weak, and not yet suitable as a basis for operational models. It is however worth noting that the operational method used for extraction of $f_{MP}$ from MODIS has $R^2_{fit}$ values ranging between 0.28 and 0.45, not too far from our values of 0.21 and 0.26. We also emphasize that we do not intend to develop an operational model based on a

single SAR scene. To differentiate this we have changed the wording "model" to "regression", "regression fit", or ''estimation'' in the manuscript. This modification is implemented several places in the manuscript. Hence, with improved co-location (see point g) below) and more satellite data, X-band SAR can potentially be used for $f_{MP}$ estimation. The following changes have been made in the manuscript to stress these points:

- The following sentences were added (Results, P12L363, P13L422): "This implies a weak correlation, corresponding well to Spearman's correlation of 0.45."
- The following sentence was added (Discussion, P13L417, P16L506-512): "The results of this study show that $f_{MP}$ influences the signature of several X-band polarimetric features. The strongest correlations were found for $R_{VV/HH}$ and $\sigma^0_{VV}$, where linear regression fits gave $R^2_{fit}$ values of 0.21 and 0.26, respectively. These correlations are not strong enough for the results to be used directly in operational models. However, with improved methods and more satellite data added, our results imply a future potential in retrieving $f_{MP}$ from X-band SAR. For comparison, the method developed for retrieval of $f_{MP}$ from MODIS has $R^2_{fit}$ values ranging from 0.28 to 0.45 (Rösel et al. 2012)."
- The following paragraph was rephrased (Results, P11L247-351, P13L398-403): "The melt ponds affect the polarimetric signatures in scene T3 and T4 differently (Table 4 and Fig. 4 and 5), mainly due to different wind conditions, but also due to different incidence angles and noise floors. In the following, we look closer into the feature displaying the strongest correlation to $f_{MP}$ in each of the scenes, $R_{VV/HH}$ in T3 and $\sigma^0_{VV}$ in T4."
- The following sentence was rephrased (Results, P12L360, P15L471-474): "As for the intermediate wind case, a robust least square linear fit was applied to the data to describe the relationship between $\sigma^0_{VV}$ and $f_{MP:}$"

g) From the reviewers comment, we find that the large scatter observed in the scatter plots, reflected in weak correlations values should be discussed in more detail in the manuscript. Deformation (and volume scattering in the sea ice) may contribute to the low correlation values, but another source is probably equally important. Substantial efforts were made in co-locating the helicopter photos and the floe's position. As stated in the Method-section we estimate a possible areal offset between the helicopter images and the compared corresponding SAR pixels of up to 27%. Even a small positional offset of a few percentages would introduce random noise in the regression, lowering the correlation values. Viewed against this background, we find it acceptable to explore the effect of the regression fit on the floe and the full scenes in the data set, even if the $R_{fit}^2$ values are low. The following changes have been implemented in the manuscript:

- The following sentence was added (Results, P12L363, P13L423-425): "However, the co-location between the helicopter images and the sea ice floe contain some uncertainty (a maximum areal offset of 27%) possibly introducing a random error to the regression, resulting in an artificially low $R_{fit}^2$."
- The following paragraph was added (Discussion P15L490, P18L605-614): "The correlations found in our study are not very strong. The weak to moderate correlations might suggest a limited sensitivity to $f_{MP}$ in X-band SAR imagery, but they could also reflect limitations in the data set. The co-location between the helicopter images and the SAR imagery is estimated to have a possible offset of at most 27% potentially introducing a large random error into our investigation, lowering the correlation values. A larger degree of smoothing than the area covered by the helicopter images allows for might also be needed to improve the results. The absolute radiometric accuracy of TSX scenes could also influence the results of our study, but this influence is expected to be very small compared to other uncertainties. All the above-mentioned issues should be addressed in future studies."
- The following sentence was rewritten (Conclusion, P16L515, P19L640-645): "Challenges in co-location of airborne observations and SAR imagery limited coordinated use of existing

data in our study and introduced uncertainties in our results, possibly causing artificially low correlation values."

h) We agree that the quality of the method when applied to the full satellite scenes can be hard to evaluate. To improve this, we have included the figures suggested in the reviewers point 3). We have also moderated the way we present and discuss our results in the Abstract, Discussion and Conclusion sections. Hence, the following changes are introduced:

- The following sentence was moderated (Abstract, P1L18, P1L20) "Despite this, our findings suggest new possibilities in melt pond fraction estimation from SAR, opening for expanded monitoring of melt ponds during melt season."
- The following paragraph was moderated (Discussion, P13L417-419, P16L506-607): "The results of this study show that $f_{MP}$ influences the signature of several X-band polarimetric features."
- The following sentence was moderated (Discussion, P14L435, P16L532-534): "In our study we found a significant correlation between $R_{VV/HH}$ and $f_{MP}$ at an incidence angle of 29° (T3), demonstrating that $f_{MP}$ has an impact on polarimetric X-band SAR signatures also at lower incidence angles."
- The following sentence was added (Discussion, P14L444, P17L542-543): "However, the different acquisition geometry observed in Fig. 1 could also play a role."
- The following sentence was rewritten (Discussion, P15L468, 17L568-569): "A larger window size reduces the amount of speckle in the SAR scenes, which possibly explains the improvement."
- The following sentence was rewritten (Discussion, P15L475, P17L574-576) "The large sample variability observed in Fig.4 might therefore be negligible, as long as the $R_{VV/HH}$-based regression fit produces a good estimate of the mean $f_{MP}$ for a larger area."
- The following sentence was moderated (Conclusion, P15L495, P19L618-620): "In this study we demonstrate statistically significant relations between $f_{MP}$ and several polarimetric SAR features on drifting FYI in X-band, based on helicopter-borne images of the sea ice surface combined with four dual polarimetric SAR scenes."
- The following sentences were moderated (Conclusion, P16L497, P19L621-625): "The study reveals a prospective potential for $f_{MP}$ estimation from X-band SAR, but also stresses the importance of including wind speed and incidence angle in a future robust $f_{MP}$ retrieval algorithm. Such an algorithm could supplement optical methods, and be used as a tool in climate applications, both as input in climate models and in studies of melt pond evolution mechanisms."
- The following paragraph was moderated (Conclusion, P16L505-512, P19L627-636): "The theoretical range of suitable wind speeds (<5 m/s) and sea ice surface roughnesses ($s_{RMS}$<1.4 mm) for $f_{MP}$ extraction based on $R_{VV/HH}$ are slightly more limited in X-band than in C-band but our results show that $f_{MP}$ also influences the X-band SAR signature when these criteria are partly exceeded. The high noise floor of TerraSAR-X also restricted use of scenes with incidence angles above 40°, while an incidence angle of 29° gave better results. At very low wind speeds (0.6 m/s), the backscatter signal from the melt ponds became too low for $f_{MP}$ retrieval based on polarimetric features. In that case, $\sigma^0_{VV}$ was found suitable for $f_{MP}$ estimation. In the future, use of X-band scenes can possibly increase the total amount of SAR data accessible for $f_{MP}$ retrieval, despite their limitations compared to C-band scenes."
- The following sentence was added (Conclusion, P16L525, P20L651-653): "For development of a robust operational method, future studies should aim to include a larger number of satellite scenes acquired during various sea ice conditions, melt pond evolution stages, wind speeds and incidence angles."

a) The authors compare MPF distributions from airborne and retrieved from SAR data. To evaluate the quality of the results even better, it would be good to show also the spatial situation: the MPF retrieved from SAR plotted on a lat-lon map and the airborne reference data overplotted on the same map using same colorscale. Upon checking spatial features or spatial uniformity, the reader can make sure that the retrieved MPFs are not random numbers, but really correspond to the field situation.

We appreciate the suggestion of a spatial plot. We have now included spatial MPF figures in the manuscript. The following text and Figures are introduced to the manuscript:

- The following paragraph was added (Results, P12L382, P13L445-455): "Zooming in to the southern part of the area covered by the helicopter survey on the floe in T3, Fig. 6 *(new)* displays $f_{MP}$ estimated from Eq. 17 with the observed $f_{MP}$ from the helicopter images overlaid. Two different pixels smoothing windows are shown (21 x21 and 51 x 51). Note that the center pixel underlying each helicopter image frame would give the most representative value for comparison to the observed $f_{MP}$, as pixels closer to the frame contain a larger amount of information from outside the frame. The middle panel displays the mean estimated $f_{MP}$ value for each frame together with the observed $f_{MP}$ values along the track. The maps confirm some overlap between the estimated and observed $f_{MP}$, but also illustrates that there is room for improvement. The estimation with a 51 x51 pixel smoothing window appears less variegated than the 21 x 21 estimation, and the range of the estimated $f_{MP}$ values also corresponds better to those observed from the helicopter images in the 51 x51 estimation."

- The following paragraph was added (Results, P13L410, P15L488-497): "Figure 10 *(new)* shows $f_{MP}$ estimated from Eq. 18 with the observed $f_{MP}$ from the helicopter images overlaid for two different pixels smoothing windows (21 x 21 and 51 x 51). Note that the center pixel underlying each helicopter image frame would give the most representative value for comparison to the observed $f_{MP}$. To illustrate this, the middle panel shows the mean estimated $f_{MP}$ value for each frame together with the observed $f_{MP}$ values along the track. In general, a good overlap between the estimated and observed $f_{MP}$ can be seen, even though some scatter exists. As in Fig. 6 the estimation with a 51 x 51 pixel smoothing window appears less variegated than the 21 x 21 estimation, and the range of the estimated $f_{MP}$ values also corresponds better to those observed from the helicopter images in the 51 x 51 estimation than to those in the 21 x 21 estimation."

[Figure]

Figure caption: "Figure 6. Melt pond fraction ($f_{MP}$) estimated from $R_{VV/HH}$, with the observed $f_{MP}$ from the helicopter images overlaid as colored frames. The area displayed is outlined with a frame in Fig. 5. The estimation is performed with 21 x21 (left) and 51 x51 (right) pixels windows. Note that the center pixel underlying each helicopter image frame would give the most representative value for comparison to the observed $f_{MP}$, as pixels closer to the frame contain a larger amount of information from outside the frame. The middle panel displays the mean estimated $f_{MP}$ value for each frame together with the observed value."

[Figure]

Figure caption: "Figure 10. Melt pond fraction ($f_{MP}$) estimated from $\sigma_{VV}$, with the observed $f_{MP}$ from the helicopter images overlaid as colored frames. The area displayed is outlined with a frame in Fig. 8. The estimation is performed with 21x21 (left) and 51 x 51 (right) pixels windows. Note that the center pixel underlying each helicopter image frame would give the most representative value for

comparison to the observed $f_{MP}$ as pixels closer to the frame contain a larger amount of information from outside the frame. The middle panel displays the mean estimated $f_{MP}$ value for each frame together with the observed value."

The authors come to the conclusion that the dual polarimetric SAR data in X band can be used for melt pond estimate given the appropriate wind speed, incidence angle, surface deformation ranges and also upon extensive smoothing or even taking the mean value over the whole scene.

The impact and importance of such a product is not sufficient for advancing our under-standing on melt pond processes and can only serve as complementary data for other studies. Currently, the manuscript serves more as a fundamental study on the SAR features in X band and more displays limitations than advantages of the data.

We agree that the manuscript can be seen as a fundamental study on polarimetric SAR features in X-band, and their relation to melt pond fraction, testing a potential method for melt pond fraction retrieval. Producing an operational product for estimation of melt pond fraction from (X-band) SAR will take more than one single study. To advance in this process, it is very important to focus on possible limitations like surface roughness ranges and wind speed, and optimum SAR parameters like incidence angle and smoothing window size (stated in Conclusion, P15L522-529, P19649-654). This allows for more precise and to the point studies in the future. Hence, we think our study presents important results for future development of melt pond fraction retrieval from X-band SAR.

I recommend to support the shown MPF results with possibly more SAR scenes and definitely show the spatial MPF maps to confirm the quality of the pond retrieval, or refocus the manuscript on signatures of various ice/pond types in X band without the actual MPF retrieval.

SAR-scenes with corresponding ground truth are very rare, being generally a result of coordinated campaigns, and in this data set, we are limited to the presented scenes. As detailed above, we have included the requested MPF maps (new Figs 6 and 10), allowing the reader to confirm the quality of the presented method.

Technical
- Please add the error bar of the empirical fit in Eq. (16) and (17) on the corresponding figures, this helps to estimate the quality of the MPF retrieval.

95% confidence intervals were added in Figure 4 and 7.

- please add the correlation coefficient values into the abstract and into figure captions where you present the empirical fits.

Correlation coefficient values were added into the figure captions of Figure 4 and 7, and into the abstract:
- The following sentence was rephrased (Abstract P1L10, P1L9-13): "To further investigate these relations, regression fits were made both for the intermediate ($R^2_{fit}$ =0.21) and low ($R^2_{fit}$ =0.26) wind case, and the fits were tested on the satellite scenes in the study."

- I suggest to merge the subsection 4.1 Sea ice conditions into the subsection 3.1 Study region. Current section 4.1 logically fits better to 3.1.

The subsections have been merged, and are now united in subsection 3.1 renamed "Study region and sea ice conditions".

---

## Author Comment (AC2) · 25 Nov 2016

Response to review by reviewer #2

We thank the reviewer for constructive criticism, allowing us to improve our manuscript. A majority of the reviewer's requests have been met. Below follow our answers and comments, highlighted in blue after each of the reviewer's comments. Blue page (P) and line (L) numbers refers to the manuscript published in TCD and red numbers refers to the marked-up revised version.

**General comments**

The authors have studied melt pond fraction (MPF) estimation using TerraSAR-X dual-polarization SAR imagery acquired over drift ice north of Svalbard, and presented empirical models for MPF estimation in two different wind speed conditions (low speed and intermediate speed).

In the Introduction Section authors give good overview on importance of melt ponds on the Arctic sea ice heat budget and in the Arctic climate system, and on previous studies on melt pond fraction estimation with optical and SAR imagery. The number of previous SAR studies on melt pond detection and fraction estimation is quite large, and so far a generic method for the estimation has not been presented/developed. There has been some success for the melt pond fraction (MPF) estimation over smooth landfast ice using C-band co-polarization ratio (HH and VV pol SAR images needed) or HH-pol backscattering coefficient (σ). For MPF estimation over drift ice only few studies has been conducted. At least it seems that MPF estimation over drift ice with C-band single pol imagery is not possible. Over drift ice sea ice deformation features like ice ridges and make MPF estimation in theory much more difficult than over smooth landfast ice. Other frequencies than C-band have been used only in few case studies. Likely (to my opinion) accurate MPF estimation is only possible with high resolution (<5-10 m) SAR imagery. So far time series of MPF maps over the Arctic have been produced only with optical imagery (MODIS, MERIS). These charts are limited by accuracy of automatic cloud masking and persistent cloud cover during the Arctic summer. Accurate MPF charts from SAR imagery would supplement greatly the MPF charts from optical imagery.

Section 2 discusses nicely about melt pond signatures in SAR imagery, but it could also include overview of observed MPF behavior during melt ponding season (ponding, drainage etc.), see e.g. D. G. Barber, J. J. Yackel, and J. M. Hanesiak, "Sea ice, RADARSAT-1 and arctic climate processes: A review and update," Can. J. Remote Sens., vol. 27, no. 1, pp. 51–61, 2001.

We appreciate this suggestion, and have included a brief comment on this in the manuscript.
- The following sentences were added (Melt ponds in SAR imagery, P4L113, P4L127-130): The coverage of melt ponds varies during the melt season, starting out with a high fractional cover, and reducing as the ponds drain. At the end of the melt season, the melt ponds refreeze. This evolution is mirrored by a seasonal variation in the sea ice SAR signature (Barber et. al 2001)."

Dual-polarization TerraSAR-X imagery acquired over drift ice for MPF estimation have been previously studied by Kim et al. (2013) and Han et al. (2016). Kim et al. (2013) used only one TSX image acquired in Aug 2011 over East Siberian Sea, but they have large amount of co-incident airborne very fine resolution X-band (single pol) images. Comparison MPF data was from airborne photography. Han et al. (2016) used the same datasets and also one additional TSX image acquired in July 2011 over the Chukchi Sea. Kim et al. (2013) estimated MPF with "We first delineate the ice and melt pond features using image processing software (ENVIA EX), based on the combination of multiscale segmentation and aggregation methods."; not discussed in more details, but Han et al. (2016) studied various polarimetric parameters and their textural features in MPF estimation by machine learning approaches.

The authors have used here four TSX dual-polarized StripMap images acquired during ICE2012 campaign in north of Svalbard. Comparison MPF data was from helicopter-borne optical imagery. In addition, surface roughness data was calculated from stereo camera imagery, and weather data was measured by R/V Lance at the ICE2012 campaign site. They have studied MPF estimation with different polarimetric parameters calculated from the dual-pol TSX imagery, as was done also by Han et al. (2016). The main questions now are: 1) Does this study give new scientific results/information compared to Kim et al. (2013) and Han et al. (2016)? 2) In what ways it is different to them, data and/or methods?

My answers: Study area is different (Chukchi Sea vs. Svalbard) which could have influence on the results if sea ice conditions (FYI vs. MYI) where different; authors should discuss this in the paper. Wind speed is taken into account here unlike in Kim et al. (2013) and Han et al. (2016). Wind speed has large effect on the backscattering from melt ponds (not frozen). Somewhat more SAR images have used, four here compared to two in Han et al. (2016), making the results here more reliable. The developed MPF algorithms are linear functions between MPF and one polarimetric parameter. I favor this kind of simple approach as the results can be related easily to theoretical backscattering models. Han et al. (2016) utilized machine learning approaches where relations between polarimetric parameters (and scattering theory) and an estimated parameter may not be very clear. However, I think that the paper in its current form gives quite little new information/findings compared to previous studies on the MPF estimation with SAR.

We appreciate the papers by Kim et al. (2013) and Han et al. (2016), which as our study focus on melt ponds and SAR. Kim et al. (2013) focuses on airborne SAR and barely discuss satellite imaging and is hence very different from our study. Han et al. (2016) presents a machine learning approach for melt pond fraction retrieval from X-band SAR based on two satellite SAR scenes, one that has corresponding information about actual melt pond fraction.

As the reviewer has commented on, there are several differences between the study of Han et al. (2016) and our study. Han et al. (2016) focuses on MYI, and explicit request more studies on other sea ice types in their conclusion. Our manuscript focuses on level FYI, which has different microwave signature than MYI, as commented on in Han et al. (2016). The effect of sea ice surface roughness, satellite noise floor, wind speed, and incidence angle are not discussed in Han et al. (2016), but wind speed and incidence angle are suggested to be investigated in future studies. These are important factors in understanding the melt pond signature of X-band SAR and are therefore included in our study, adding essential information. Only one scene with corresponding melt pond information was employed by Han et al. (2016. Increasing the number of scenes, as done in our manuscript, is of large importance to improve the understanding of melt ponds signature in SAR imagery.

While Han et al. (2016) focuses on combining several polarimetric signatures in machine learning algorithms for melt pond retrieval, our manuscript concentrates about the individual polarimetric features, and the influence of melt ponds on these. This could form a basis for more advanced methods later on, guiding which features to use under specific wind speed conditions and incidence angles. Melt ponds and sea ice are treated as different classes in the study of Han et al. (2016). Our study on the other hand, focuses on the signature of mixtures of melt ponds and sea ice.

When it comes to the results, different polarimetric features are found sensitive to melt pond fraction in Han et al. (2016) and our study. While co-polarisation ratio and HH intensity were the most promising features in our study, co-pol phase difference, alpha angle, and HH intensity were found to be the most important ones in Han et al (2016). This could be due to e.g. difference in wind conditions, which there is little information about in the study by Han et al. (2016).

In summary, we find that Han et. al (2016) and our study does not present much overlapping information and findings, hence our study are complementary. The sea ice types, methods and results are different, and our study includes several factors (more scenes, wind speed, incidence angle, surface roughness etc.) not considered in Han et al. (2016). To evolve in the understanding of melt ponds signatures and impact on X-band SAR, and SAR in general, a variety of studies is necessary, and we believe our study contributes with enough new insight, and is worthy of publication.

To clarify the differences in the study of Han et al. (2016) and our study for the readers of our manuscript, the following changes have been applied:

- The following sentences were rephrased (Introduction, P3L82, P3L85-91): "Han et al. (2016) combined multiple polarimetric SAR features in MPF estimation by machine learning methods, employing the co-polarisation channels of the MYI X-band SAR scene explored in Kim et al. (2013). An additional scene was also included in the study, though without melt pond information. The study showed promising results, but the authors claim that more scenes with various sea ice types and incidence angles are needed to develop a general propose MPF model. Lack of wind information is also limiting the relevance of the study."

- The following paragraph was added (Discussion, P15L490, P18L592-604): "The findings in our study deviate from the findings of Han et al. (2016) where $\sigma^{o}_{HH}$, $< \rho$, and, $\alpha_1$ were found to be the most prominent polarimetric features in separating melt ponds, sea ice and open water in high resolution X-band SAR imagery. Differences in sea ice type, sea ice surface roughness, wind conditions, and SAR incidence angle could possibly explain why different polarimetric features are sensitive to MPF in the two studies. The methods of the two studies are also slightly different, as Han et al. (2016) classify each pixel into melt pond, sea ice or open water, while our study focuses on mixtures of melt ponds and sea ice. Exact wind information lacks in Han et al. (2016) but the wind speed is expected to be low. This could explain why $\sigma^{o}_{HH}$ contributes strongly in MPF estimation, and is then in accordance to our findings. The diverging results in the two studies emphasize the need of investigating melt ponds impact on SAR imagery under different conditions and for a variety of sea ice types. It also stresses the importance of supplementary measurements of parameters like wind speed and sea ice surface roughness."

The statistical reliability of the developed empirical MPF estimation models seems quite low, r2 was at best only 0.21 and RMSE is high.

We agree that the $R^2$ values for the regression fits are low. For our comment on this, see point 2f) in the reply to Reviewer #1.

The value of the paper could be improved by following changes and additions:

The empirical models for the MPF estimation were developed using datasets over a large ice floe. Why were not all co-incident SAR imagery vs. airborne photography used? How results would change if they were?

As this study was performed on drifting sea ice, co-location between SAR scenes and helicopter data is very challenging. This is also commented on in the conclusion of the manuscript. Most of the airborne photos were not possible to be co-located with the satellite observations exact enough to meet our demands of a high quality study. However, for the investigated floe, we managed to do a reliable co-location, and the floe was also the only floe appearing in two of the scenes. We therefore chose to focus on this specific floe in our investigations to secure the quality of the study.

Both wind speed and SAR incidence angle have large effect on the MPF estimation. Wind speed is now taken into account by MPF models for two different wind conditions. I suggest you developed

MPF models which include incidence angle or compensate σ incidence angle variation before MPF estimation. The study should include more variable wind speed conditions, but in the current dataset these are not present

We agree that it is desirable to study the melt pond signature under as many wind speed situations as possible to make a robust melt pond estimation. Our data set consists of four scenes, and it is therefore not possible to make such a robust model from this small dataset. However, our study highlights the importance of wind speed, and can serve as a starting point for future studies. This is already commented on in the conclusion of the manuscript.

Incidence angle correction has been introduced to the manuscript, to improve the understanding of Fig. 9. The following changes have been made in the manuscript:
- The following sentences were added (Method, P10L295, P11L340-346): " Incidence angle correction was applied to the scenes for a better comparison, employing the following equation (Kellndorfer et al., 1998)
$$\sigma°_{corr}= \sigma° \, (\sin θ/ \sin θ_{ref})$$
where $\sigma°$ is the original backscatter coefficient, $θ$ is the center incidence angle of the scene to be corrected, and $θ_{ref}$ is the reference incidence angle of scene T4. The correction was only applied in the low-wind case, as it canceled in the intermediate wind case due to the use of a co-polarisation ratio."
- The following reference was added: Kellndorfer, J., L.E., P., M.C., 715 D., and Ulaby, F. T.: Toward consistent regional-to-global-scale vegetation characterization using orbital SAR systems, IEEE Transactions on Geoscience and Remote Sensing, 36, 1396-1411, doi:10.1109/36.718844, 1998.
- The following sentence was rewritten (Results, P13L413, P15L501-502): "Incidence angle correction according to Eq. 16 is applied to the figure, accounting for $\sigma°_{VV}$ decrease with incidence angle."
- The following sentence was rewritten (Discussion, P14L449, P17L548-550): "The underestimation of FMP in scenes T1-T3 is likely related to higher wind speeds at the time of acquisition."

You could study effect of sea ice type in the MPF estimation, e.g. by first segmenting the SAR images to level ice and deformed ice categories (with the help aerial photography if possible). In best case we could have also sea ice type taken into account in the MPF estimation. You have also surface roughness data which could be utilized here.

We agree that the effect of surface roughness could be better presented in the manuscript. We have now introduced two classes of sea ice in the scatter plots (Fig. 4 and 7), representing totally level ice and partly deformed sea ice. The classification is based on visual interpretation of the helicopter images. The following changes have been introduced in the text in relation with the classification:
- The following sentences were added (Results, P11L355, P13L408-411): Grey dots correspond to areas with some degree of sea ice deformation, while blue dots correspond to areas with completely level ice. Deformation information is extracted from visual inspection of the helicopter images."
- The following sentence was rewritten (Results, P12L358, P13L414-416): "A majority of the lowest $R_{VV/HH}$ values are appearing in partly deformed areas. Areas with some degree of deformation also represent the lowest $f_{MP}$."
- The following sentence was added (Results, P13L395, P15L470-471): "Grey dots correspond to partly deformed areas, while blue dots represent level ice."

Show MPF maps from some SAR images and discuss spatial variation present, does it make sense? You have four SAR images, how does estimated MPF behave temporally? Now Table 5 shows MPF averages over the full scenes, but these are not much discussed in the text, and temporal variation does not seem right (36.2->45.7->31.2->53.3).

Spatial MPF maps have been introduced to the manuscript. See comments to point 3) in the reply to Reviewer#1.

During the campaign, the MPF was stable with a mean of 33.2%. The variation seen in Table 5 is due to differences in wind speed conditions and incidence angle. This is thoroughly commented on and discussed in the last paragraph of section 4.1 (New 3.1) and in the third paragraph in section 5 (Discussion).

Can you compare your estimates with those from optical imagery? See http://icdc.zmaw.de/1/daten/cryosphere/arctic-meltponds.html

The suggested data set lasts until 2011, and can therefore not be used in our study. We agree that it would be interesting to compare our data to optical data, and investigated this opportunity early in the manuscript process. However, there were cloudy conditions during the campaign, and optical data were therefore not accessible. This underlines the advantage of a possible melt pond fraction estimation from SAR.

The study would benefit greatly from a larger SAR dataset. Are there any co-incident TSX vs. in-situ / airborne data from NICE2015 campaign you could use? You really need more wind speed conditions for the MPF estimation development. Even including more TSX images without corresponding comparison data is possible, you could study spatial and temporal trends. In addition, any fine resolution C-band images available? Comparison between C- and X-band would be nice addition.

We agree that one should aim to use as many SAR scenes as possible in a study like this. But this has to be balanced by the actual access to scenes with high quality information about melt pond fraction and sea ice conditions retrieved from in situ measurements and helicopter photos. Such data sets are very rare, and it is therefore important to publish results from existing data sets, even if they as in our case have a limited number of scenes.

The N-ICE-2015 campaign was finished in mid-June 2015, before the onset of intense surface melt, and data from this campaign is therefore not appropriate for our study. C-band scenes were planned collected during the campaign, but acquisition priorities hampered collection of such scenes. We have also attended other campaigns with intention of increasing the data amount. The lack of success with these efforts emphasizes the value of the presented data set.

In general, the paper is well written and structured, and easy to read and understand. The data processing and analysis methods are scientifically sound and discussed in needed detail. I am afraid in the current form the paper gives quite few new scientific findings compared to previous studies.

We appreciate that the reviewer likes our manuscript. The novelty of this study is already discussed above. We find that our study brings in new and different findings compared to the only other existing study on the topic (Han et al. 2016), and we believe it is worthy of publication.

From Conclusions: "Future studies should aim to include a larger number of satellite scenes acquired during various sea ice conditions, melt pond evolution stages, wind speeds and incidence angles. The effect and limitations of sea ice surface roughness and dependency on filtering size and scale should also be further investigated."

You should consider taking some of these topics to this paper!

Wind speed, incidence angle, surface roughness, and filtering size are all discussed in our manuscript. This sentence simply states that more data is needed to make a robust algorithm for melt pond fraction retrieval in X-band. To make this absolutely clear, we have rewritten the sentence:
- The following sentences were rewritten (Conclusion, P16L525, P19L651-654): "For development of a robust operational method, future studies should aim to include a larger number of satellite scenes acquired with various sea ice conditions, melt pond evolution stages, wind speeds, and incidence angles. The effect and limitations of sea ice surface roughness and dependency on filtering size and scale should also be further investigated."

Finally, Yackel and Barber (2000) speculated that MPF may be more closely related to the albedo than to melt pond fraction due to the fact that albedo results from the integration of all surface types (snow, saturated snow, melt ponds) which contribute to the measured MPF. What's the authors' view on this; would it be better to investigate the relationship between SAR data and albedo than SAR and melt pond fraction? Please, discuss this in Introduction Section.

This question might have a typing error, and we find it slightly unclear. We interpret the question to ask whether estimated melt pond fraction should be compared to albedo instead of observed melt pond fraction. Albedo refers to the average reflection of waves in the visible range of the microwave spectrum. As SAR uses microwaves to evaluate the sea ice surface, we find it more credible to utilize differences in the microwave signature between melt ponds and sea ice, or methods that employ statistical features describing fractional mixtures of surfaces.

In our study, albedo is not measured, and would have to be estimated by upscaling from melt pond fraction measurements and in situ measured albedo values of different surface types. This method would inevitably introduce additional uncertainty to the results (see f.ex. Divine et al., 2015 for estimates made for the study area), and is therefore not advisable for our data set. The relationship between albedo and polarimetric features will therefore not be discussed in detail in our manuscript.

**Specific comments**
**1.Introduction**
page 3, lines 90-92: terms 'dual polarimetric' and 'dual-polarisation' used, confusing…I think it should be 'dual-polarisation' for SAR imagery with two polarizations.

We agree in this, and have changed the term:
- The following sentence was rewritten (Introduction, P3L90-92, P4L101-103): "The objective of this study is to investigate the potential of melt pond fraction retrieval from level drifting FYI with dual-polarisation X-band satellite SAR. A data set consisting of four high resolution dual-polarisation TerraSAR-X satellite scenes…"

**2. Melt ponds in SAR imagery**
p. 4, l. 118: "Observed surface roughness increases with increasing frequency, making X-band more sensitive to small-scale surface roughness than C-band."
I think surface roughness is physical property of a surface, and its effect on backscattering depends on radar wavelength.

We agree that this sentence was unprecise, and have rewritten it:
- The following sentence was rewritten, (Melt ponds in SAR imagery, P4L118, P5L135-137): "X-band is more sensitive to small-scale surface roughness than C-band, as the effect of surface roughness depends on radar wavelength."

l. 124: "Six of these features are included in our study and are described in the following subsection."
In Table 4 there are eight features.
This is correct, and we have corrected the sentence:
- The following sentence was rewritten (Melt ponds in SAR imagery, P4L124, P5L142): "Eight of these features..."

**3.2 Data set**
Give absolute calibration accuracy and equivalent number of looks (ENL) (i.e. effect of radar fading) in TSX images. Do they have any significant effect on your data analysis results?

The absolute radiometric calibration accuracy is 0.6 dB. This parameter is an image-wide measure and includes temporal drift. Hence, it probably varies more from near-range to far-range than in a local region. As we focus on small regions in our main analysis, this accuracy will probably not play a significant role. However, for development of a future operational algorithm across many images, the calibration accuracy might be of importance. We therefore include the accuracy in our manuscript, and note that it should be further explored in future studies. The following changes were introduced in the manuscript:
- The following sentence was added (Method, P8L231, P9L272): "The absolute radiometric calibration accuracy of TSX is 0.6 dB (Airbus Defense and Space, 2013)."
- The following sentence was added (Discussion, P15L490, P18L611-614) "The absolute radiometric accuracy of TSX scenes could also influence the results of our study, but this influence is expected to be very small compared to other uncertainties."

ENL is a far more complicated issue, and we do not see how including this would strengthen the manuscript. We are not statistically modelling the speckle distributions, and only look at the mean values after smoothing. ENL does have some bearing on the degree of variation, but this is directly evident form our observations without the complication of trying to interpret the effect of ENL. Also, radar texture, or non-Gaussianity, which is observed to be high for TSX imagery, is a far more influential factor. All simple ENL measures are Gaussian-based estimates, which do not correctly capture the texture aspect, and could lead to incorrect interpretation. Based on these reflections, we have decided not to include ENL in the manuscript.

**4.1 Sea ice conditions**
Was there any nighttime re-freezing on sea ice and melt pond surfaces which could have influenced backscattering signatures in T1 TSX image acquired at 06:52 UTC on 28 July?

Nighttime refreezing was not observed during the campaign, and is therefore unlikely to have influenced any of the SAR signatures. This is now clarified in the manuscript:
- The following sentences were changed/added (Method, P10L314, P8L256-257): "Air temperatures were varied little between -1 to 1.5C. Combined with the oceanic heat flux, the ice was therefore in continuous melt even at nighttime."

**4.3 Intermediate-wind case**
p. 12, l. 358: "From visual inspection of the helicopter images, some of the lowest RVV/HH values origin from slightly deformed areas with a surface roughness possibly exceeding the Bragg criterion." Please discuss how this sea ice condition leads to low RVV/HH.

We agree that this should be better described. We have therefore expanded the sentence with a possible explanation of the low values.

- The following sentence was rewritten (Results, P12L358, P13L411-414): "The partly negative values of RVV/HH imply that $\sigma^0_{HH} > \sigma^0_{HH}$. This might be a result of multiple scattering events in the sea ice volume or sea ice surface, possibly connected with sea ice deformation."
- The following sentence was rewritten (Discussion, P14L460, P17L559-560): "Multiple scattering events in the sea ice surface and sea ice volume may also have contributed to the large sample variations observed in Figs. 4 and 8.

---

## Referee Report (RR1)

**Signature of Arctic first-year ice melt pond fraction in X-band SAR imagery**

Ane S. Fors, Dmitry V. Divine, Anthony P. Doulgeris, Angelika H. H. Renner, and Sebastian Gerland

Manuscript prepared for The Cryosphere, Date: 25 November 2016

**Review of the revised paper**

**General comments**

The authors have presented detailed, proper answers to all my comments to the previous version of the paper, and made corresponding changes and additions to the paper. I think that the paper has improved considerably. Below are some further comments, the major one is related to the T4 data and its analysis.

The empirical models for the MPF estimation were developed using datasets over a large ice floe. Why were not all co-incident SAR imagery vs. airborne photography used? How results would change if they were?

As this study was performed on drifting sea ice, co-location between SAR scenes and helicopter data is very challenging. This is also commented on in the conclusion of the manuscript. Most of the airborne photos were not possible to be co-located with the satellite observations exact enough to meet our demands of a high quality study. However, for the investigated floe, we managed to do a reliable co-location, and the floe was also the only floe appearing in two of the scenes. We therefore chose to focus on this specific floe in our investigations to secure the quality of the study.

You could also explain this shortly in the paper.

Results Section could include short introductory at the start about its content.

Reference Divine et al. (2016, in review) describes the method to estimate DTM and surface roughness from stereo camera photos, but as its 'in review' readers do have access to it, any published references? Conference papers? I have to say I don't personally like these 'in review' references, as they may not get published in the end.

ENL is a far more complicated issue, and we do not see how including this would strengthen the manuscript. We are not statistically modelling the speckle distributions, and only look at the mean values after smoothing.

I was thinking about radiometric resolution: $10*\log10(1+1/\sqrt{ENL})$, but it is likely insignificant for mean sigma0 values. I have estimated ENL over textureless open water areas. Your comment on the subject was ok.

Section 3.2 Data set: line 272. Reader gets an idea that NESZ was subtracted from SAR imagery, and then images are analyzed, but later Section 4.1 results are presented with and without noise reduction in SAR imagery. I think you should selected either way, or does having both add great value to your results and paper?

Section 4.1, lines 386-397 and Table 4: For SAR scene T4 acquired at very low wind speed, HH and VV sigma0 decreases with increasing melt pond fraction. This makes sense as there are more low backscatter smooth melt pond surface with increasing mpf. However, RVV/HH also decreases with increasing mpf (negative Spearman's correlation in Table 4), I found this puzzling as smooth surface should have high RVV/HH. In addition, in (Scharien et al. 2014a), Figure 5, RVV/HH is not decreasing with increasing mpf. Scharien et al. had strong positive relationship between RVV/HH and mpf is low wind speed (1.1 m/s) case. In Figure 10 of Yackel et al. (2000) s0VV slightly increases with increasing mpf under low wind speed conditions (1.5 m/s). Something wrong in your SAR data, too large contamination by SAR noise? If not, how do you explain the observations?

Would be nice to see RVV/HH vs. mpf plot for the T4 case. How RVV/HH vs. mpf in T4 case changes if you set a lower signal-to-noise ratio for s0 data, for example 5dB, i.e. only those RVV/HH datapoints with SNR larger than 5 dB for s0HH and s0VV are accepted? I think you need to investigate further your T4 data.

I am should have presented this observation in my first review, sorry about this…but I noticed this now.

Finally, Yackel and Barber (2000) speculated that MPF may be more closely related to the albedo than to melt pond fraction due to the fact that albedo results from the integration of all surface types (snow, saturated snow, melt ponds) which contribute to the measured MPF. What's the authors' view on this; would it be better to investigate the relationship between SAR data and albedo than SAR and melt pond fraction? Please, discuss this in Introduction Section.

This question might have a typing error, and we find it slightly unclear. We interpret the question to ask whether estimated melt pond fraction should be compared to albedo instead of observed melt pond fraction. Albedo refers to the average reflection of waves in the visible range of the microwave spectrum. As SAR uses microwaves to evaluate the sea ice surface, we find it more credible to utilize differences in the microwave signature between melt ponds and sea ice, or methods that employ statistical features describing fractional mixtures of surfaces.
In our study, albedo is not measured, and would have to be estimated by upscaling from melt pond fraction measurements and in situ measured albedo values of different surface types. This method would inevitably introduce additional uncertainty to the results (see f.ex. Divine et al., 2015 for estimates made for the study area), and is therefore not advisable for our data set. The relationship between albedo and polarimetric features will therefore not be discussed in detail in our manuscript.

My idea was that as backscatter measured with SAR, if pixel size is not too small, is a mixture of backscatter from meltponds, bare ice, snow covered ice, then the backscatter could be more related to albedo than melt pond fraction. Albedo is naturally derivative of mpf. This is what Yackel and Barber (2000) speculated, see page 22068, left column. But you can leave this out the paper.

**Specific comments**

**Abstract**

"The regression fits gave good estimates of mean melt pond fraction for the full satellite scenes, deviating with less than 4% from the airborne retrieved melt 15 pond fractions in the investigated area."

Mention shortly the airborne data used in the melt pond fraction estimation.

"A smoothing window of 51x51 pixels gave the best reproduction of the width of the melt pond fraction distribution."

Give also pixel size of SAR images, only window size in pixels does not tell geometric size on the area.

"optical satellites"

better "optical imagery"

**1 Introduction**

l. 36: "Formation and evolution of melt ponds are poorly represented in sea ice models, potentially contributing to an underestimation of the observed sea ice extent reduction in model projections (Flocco et al., 2012; Holland et al., 2012; Flocco et al., 2015)."

Possible to explain shortly the reason for this underestimation? If not, then there are these three refs for the readers.

l. 42: "Several algorithms have been developed for retrieval of melt pond fraction…"

$f_{MP}$ defined before, could be used instead of "melt pond fraction". Check also whole text for the same issue.

l. 104: "TerraSAR-X offers very high resolution multi-polarimetric data,…"

What do you mean by multi-polarimetric data? You have dual-polarisation TSX SAR images.

You could add at the end of Introduction Section short overview of the content paper Sections. Like "First, Section 2 presents …"

**2 Melt ponds in SAR imagery**

l. 118: "During very calm conditions, the SAR signal of melt ponds is mainly specular."

Maybe better "scattering from melt ponds is mainly specular"

p. 3, l. 127: "The coverage of melt ponds varies during the melt season, starting out with a high fractional cover, and reducing as the ponds drains. At the end of the melt season, the melt ponds refreezes."

I think melt pond fraction first increases, has a maximum level, and then decreases, ending up with remaining ponds refreezing, see e.g Fig. 6 in Rösel, A., Kaleschke, L., and Birnbaum, G.: Melt ponds on Arctic sea ice determined from MODIS satellite data using an artificial neural network, The Cryosphere, 6, 431–446, doi:10.5194/tc-6-431-2012, 2012.

l. 184: "The sea ice surface roughness was found to high to fill the criterion…"

'too high'

l. 201: "Entropy (H) is a part of the H=A= polarimetric decomposition, based on the eigenvectors and eigenvalues of T , describing SAR scattering mechanisms."

This sentence should have a reference to the polarimetric decomposition.

**3 Methods**

Figure 2 in grayscales could show better sigma0 variation.

**4 Results**

l. 375: "and values in parentheses show results after before NESZ subtraction of the signal"

…NESZ subtraction in the SAR imagery…

Figure 4: gray and blue dots does not separate well; very long minus sign on -0.49

Figure 6 and 10: what is the spatial size of the images in km?

**5 Discussion**

l. 558 "From the helicopter images, some of the very low RVV/HH values observed at the investigated floe in scene T3 were from slightly deformed areas, possibly explaining the negative ratios. However, no general trend in low RVV/HH values in deformed areas was found in our study."

Paper: Sea ice type and open water discrimination using dual co-polarized C-band SAR, T Geldsetzer and J J Yackel, Canadian Journal of Remote Sensing Vol. 35 , Iss. 1,2009

may have intetesting results in the context of your paper. Average RVV/HH larger than 0 dB for various ice types, but <0 dB observations also exist.

In Figure 4 large amount of negative RVV/HH values, not saying they are wrong, but kinda puzzling…calibration error between TSX HH and VV channels? If so, I guess there is nothing you can do the correct it.

**6 Conclusions**

l. 632: "At very low wind speeds (0:6 m/s), the backscatter signal from the melt ponds became too low for fMP retrieval based on polarimetric features. In that case, s0VV was found suitable for fMP estimation."

If SNR is too low for polarimetric features, then why it is not too low for s0? Again, the T4 data may be too contiminated with SAR noise for mpf retrieval studies.

---

## Author Response (AR2)

We thank the editor for useful comments and questions, below follows our answers and changes, highlighted in blue after the editor's comments. Blue page (P) and line (L) numbers refer to the revised manuscript (2nd revision), and red numbers refer to the marked-up revised manuscript.

A general question/concern for me is the suitability of X-band for the melt pond application. Could the results be used to argue for future X-band missions? Or specific modes? Or should space agencies better invest in C/L-band, perhaps conically scanning to avoid the effect of incidence angle dependency and reduce the wind speed dependency?

This is an important question. Our study indicates a potential for melt pond fraction retrieval from X-band SAR, but it does not prove that this is possible, as we do not present a ready-to-go operational method. However, if X-band is to be used for melt pond retrieval, our study give some guidance in choice of polarimetric channels, incidence angles, mode and wind speeds suitable for melt pond fraction retrieval. According to our findings, the combination of VV and HH channel is useful for melt pond fraction retrieval when wind is present, the incidence angle should preferably be above 30° but below 40°, and in our case StripMap mode, giving a high resolution, were used. The latter does not exclude other modes from being useful; this has to be further investigated. X-band does not seem to have an advantage over C-band, as X-band retrieval theoretically can be done in a narrower wind speed range, and the low noise floor of TerraSAR-X imply a disadvantage. Melt pond fraction retrieval from L-band has not yet been documented. Most of these points are summarized in the second paragraph in the conclusion, but due to the editors questions, we have added the following sentences:

- The following sentence was added (Conclusion P18L588-589, P18L606-607): "Hence, both the HH and VV polarimetric channels are needed for a future $f_{MP}$ retrieval in X-band."
- The following sentence was added (Conclusion P18L594-596, P18L612-613): "Future studies should focus on incidence angles in the range between 29° and 40°."

How was the significance calculated? I found no explanation of the method. Do you consider the independence of the data? Smoothing reduces the number of degrees of freedom and increases autocorrelation. How was this taken into consideration?

Consideration of significance was lacking from the Method section, but has now been added. Smoothing will of course increase the autocorrelation, and this could be problematic if we were working with higher order statistics. Working with mean values however, this should not be a problem that needs to be accounted for. The following sentence is now added:

- The following sentence was added (Method P11L326, P11L332): "Correlations were considered significant if they had p-values below 0.05."

A specific reviewer comment was: "Give also pixel size of SAR images, only window size in pixels does not tell geometric size on the area.". Could you please add the geometrical sizes of the windows?

Pixel spacing is already included in Table 1, and is varying in the four included satellite scenes. Hence the geometrical window size would vary slightly from scene to scene. To make this even clearer, geometrical window size is now specified in the results for T3 and T4:

- The following sentence was rephrased (Results P12L390-392, P13L401-402): "The results are presented both for a 21x 21 and a 51x 51 pixels smoothing window, corresponding to areas 50x40 m and 120x95 m in the across x along flight direction."

- The following sentence was added (Results P14L440-441, P14L452-453): "The results are presented both for a 21x 21 and a 51x 51 pixels smoothing window, corresponding to areas of 65x30 m and 155x65 m in the across x along flight direction."

I wonder how the original SAR image looks like. Usually it is very instructive to have a look at the original texture and I suggest to show the unfiltered images in comparison to the smoothed versions. Unfiltered SAR scenes would appear as noise, and we therefor do not find them useful to show. Figure 2 shows the floe in T3 and T4 with a modrate smooting of 11x11 pixels, and can be used for comparison with the extensively smoothed versions presented in Figure 5 and 9. The following change was made to make clear what smoothing was used in Fig. 2:

- Figure caption rephrased, Figure 2: The floe investigated in scene T3 (left) and T4 (right) with a 11x11 pixels smoothing window.

"Scale sensitivity was tested by using a range of different smoothing window sizes (13 × 13 to 51 × 51 pixels) in the f M P estimation". This sentence comes without a result or conclusion.
We agree that this sentence is slightly misleading and imprecise. We did not test scale sensitivity, but looked at the effect of smoothing scales. To improve this, we have made the following change:
- The following sentence was rephrased (Method P11L333-334, P11L339-340): "The effect of smoothing was tested by using a range of different averaging sliding smoothing window sizes (13 × 13 to 51 × 51 pixels) in the f M P estimation."

The unit of f_MP is confusing, given as a value beteen 0 and 1 _and_ as a percentage. I would expect values between 0 and 100 for the percentage.
This has now been corrected in all figures.

I was also confused about the definition of "empirical" and "global". A regression is an empirical technique. Perhaps use different terms?
We have changed the term "empirical" to "observed". This is done several places in the manuscript. "Global" has been changed to "region" and "regional". This is also done several places in the manuscript.

Fig. 6 looks a bit strange. The rectangles seem to have a border color? There are obvious blocky/stripe artefacts from the smoothing. Have you tried to use different window functions? I assume that you used a simple boxcar weighting but I did not found the information explicitly in the manuscript.
The rectangles represents the helicopter photos, and the border color represents the melt pond fraction observed in each photo, hence the rectangles are meant to have different border colors. A sliding averaging window (boxcar) was used in the smoothing; to specify this the following change was made:
- The following sentence was rephrased Method P11L333-334, P11L339-340): "The effect of smoothing was tested by using a range of different averaging sliding smoothing window sizes (13 × 13 to 51 × 51 pixels) in the f M P estimation."

Fig. 10 \sigma
This have been corrected.

Fig. 11: What is the meaning of negative melt pond fraction?
Negative melt pond fractions simply imply that the regression fit retrieved from T4 leads to an underestimation of the melt pond fraction in the three other satellite scene. This is commented on in the manuscript, and to be more clear about this, the following change was made:
- The following sentence was rephrased (Results P14L460-461, P15L472-473):" In the three other scenes the estimate is poor; it underestimates fmp, introducing negative fractions."

Eg. 16 use \sin
This is corrected.

20 "Despite this, our findings demonstrate suggest new possibilities in melt pond fraction estimation from SAR, opening for expanded monitoring of melt ponds during melt season. " What are the new possibilities?
21 "In the next step, melt pond estimation from SAR may supplement surveillance from optical satellites, providing melt pond information to climate applications during cloudy conditions." Isn't this a bit too optimistic? If the method would be a feasible all weather approach (still to be proven), we would need long time series and large coverage for climate applications. The application needs at the same time high SAR resolution. This could be maybe in several years ahead but certainly not for the "next step"
Due to this comment from the editor, we have moderated the two last sentences in the abstract:
- Sentences rephrased (Abstract P1L18-20, P1L20-24): "Despite this, our findings suggest new possibilities in melt pond fraction estimation from X-band SAR, opening for expanded monitoring of melt ponds during melt season in the future."

519 "Scharien et al. (2014b) finds that the Bragg criterion is exceeded for melt ponds at wind speeds above $U_{10} =\sim$ 5 m/s in X-band, reducing the expected correlation between $R_{VV}/HH$ and $f_{MP}$ above this wind speed." Sharien et al. deal with C-band. How do you justify this statement.
This statement referes to Figure 4 in Schareien et al. (2014b), that shows that melt ponds reached a rms height of 1.4 mm (the Bragg criterion) at 5 m/s wind speed. To make this more clear to the reader, the following change has been added:
- The following sentences are changed/rephrased (Discussion P15L475-478, P15L487-489): "In X-band, the Bragg criterion is exceeded for $s_{RMS}$ > 1.4 mm. Scharien et al. (2014b) finds that melt ponds exceed this roughness at wind speeds above $U_{10} =\sim$ 5 m/s, reducing the expected correlation between $R_{VV/HH}$ and fmp above this wind speed."

623 "Such an algorithm could supplement optical methods, and be used as a tool in climate applications, both as input in climate models and in studies of melt pond evolution mechanisms." Not clear to me how to use the algorithm as "input" for climate models.
We agree with the editor, and have rephrased the sentence:
- The sentence is rephrased (Conclusion P18L584-586, P18L603-604):" Such an algorithm could supplement optical methods, and be used as a tool in climate applications, e.g, in studies of melt pond evolution mechanisms."

635 "In the future, use of X-band scenes can possibly increase the total amount of SAR data accessible for f M P retrieval, despite their limitations compared to C-band scenes." Do you argue for future new X-band SAR systems? Or would C-band (or L-band) be a better choice for the application. This is not clear.

This sentence states that X-band retrieval of melt pond fraction could be a supplement to C-band retrieval in the future, but point back to the stricter limitations concerning wind speed and noise floor in X-band discussed in the preceding sentences. At this point, we have too little information to advise C-band in preference to X-band, even if our results might point in that direction.

644 "..and introduced uncertainties in our results, possibly causing artificially low correlation values" Why artificially? Fig. 2 suggests that the geo-location is not too bad?

As mentioned in the Method-section, the co-location had a maximum areal offset of 27% between the satellite images and the helicopter images. This is a considerable source of error, possibly reducing the correlation values in the manuscript. To make this clearer in the Conclusion, the following change was made:

- Sentence rephrased (Conclusion P18L603-606, P19L621-624):" Challenges in co-location of airborne observations and SAR imagery limited coordinated use of existing data in our study and introduced uncertainties in our results, with areal offsets of up to 27%, possibly causing artificially low correlation values."

Response to review of the revised manuscript by reviewer #1
We thank the reviewer for reviewing our revised manuscript, and for all useful comments provided on the improved manuscript. Below follow our answers and changes, highlighted in blue after each of the reviewer's comments. Blue page (P) and line (L) numbers refer to the revised manuscript (2nd revision), and red numbers refer to the marked-up revised manuscript.

**Review of the revised paper**

**General comments**

The authors have presented detailed, proper answers to all my comments to the previous version of the paper, and made corresponding changes and additions to the paper. I think that the paper has improved considerably. Below are some further comments, the major one is related to the T4 data and its analysis.

The empirical models for the MPF estimation were developed using datasets over a large ice floe. Why were not all co-incident SAR imagery vs. airborne photography used? How results would change if they were?

As this study was performed on drifting sea ice, co-location between SAR scenes and helicopter data is very challenging. This is also commented on in the conclusion of the manuscript. Most of the airborne photos were not possible to be co-located with the satellite observations exact enough to meet our demands of a high quality study. However, for the investigated floe, we managed to do a reliable co-location, and the floe was also the only floe appearing in two of the scenes. We therefore chose to focus on this specific floe in our investigations to secure the quality of the study.

You could also explain this shortly in the paper.

We agree in this, and have added the following sentences in the Method section:

- The following sentences were added (Method P10L301-303, P10L307-309): "This floe was chosen as it allowed for a reliable co-location between airborne images and satellite scenes, and was present in more than one scene. The rest of the airborne track was not possible to co-locate exact enough for a high-quality study."

Results Section could include short introductory at the start about its content.

To meet this request, the following sentences have been added:

- The following sentences have been added (Results P11L342-344, P11L349-351): "This section presents the results of the correlation analysis examining the relation between the investigated polarimetric SAR features and observed $f_{MP}$. It then present a brief signal-to-noise analysis, before it focuses on $f_{MP}$ retrieval in an intermediate and a low-wind case.

Reference Divine et al. (2016, in review) describes the method to estimate DTM and surface roughness from stereo camera photos, but as its 'in review' readers do have access to it, any published references? Conference papers? I have to say I don't personally like these 'in review' references, as they may not get published in the end.

We agree that "in review" references are not optimal. Divine et al (2016) is now published, and the reference was updated in the bibliography and in the manuscript.

ENL is a far more complicated issue, and we do not see how including this would strengthen the manuscript. We are not statistically modelling the speckle distributions, and only look at the mean values after smoothing.

I was thinking about radiometric resolution: 10*log10(1+1/sqrt(ENL)), but it is likely insignificant for mean sigma0 values. I have estimated ENL over textureless open water areas. Your comment on the subject was ok.

Section 3.2 Data set: line 272. Reader gets an idea that NESZ was subtracted from SAR imagery, and then images are analyzed, but later Section 4.1 results are presented with and without noise

reduction in SAR imagery. I think you should selected either way, or does having both add great value to your results and paper?

We agree in this, and have changed Table 4 to include only values retrieved after NESZ subtraction. Table caption and reference to the table in the manuscript has been updated after this change. The following changes have also been added:

- Sentences has been rephrased (Discussion P17L549-554, P17L562-569) "$\alpha\_1$ was found significantly correlated with $f_{MP}$ in scene T3. This is likely a result of the expected relation between $\alpha\_1$ and $R_{VV/HH}$ (vanZyl, 2011). In scene T4, several of the polarimetric SAR features were found related to melt pond fraction before NESZ subtraction, after NESZ subtraction, only the standard deviation of $|\rho|$ showed a relationship. This indicates that the correlations only reflected the low signal-to-noise ratio of the scene, as has previously been described in oil/water discrimination (Minchew et al. 2012)."

Section 4.1, lines 386-397 and Table 4: For SAR scene T4 acquired at very low wind speed, HH and VV sigma0 decreases with increasing melt pond fraction. This makes sense as there are more low backscatter smooth melt pond surface with increasing mpf. However, RVV/HH also decreases with increasing mpf (negative Spearman's correlation in Table 4), I found this puzzling as smooth surface should have high RVV/HH. In addition, in (Scharien et al. 2014a), Figure 5, RVV/HH is not decreasing with increasing mpf. Scharien et al. had strong positive relationship between RVV/HH and mpf is low wind speed (1.1 m/s) case.  In Figure 10 of Yackel et al. (2000) s0VV slightly increases with increasing mpf under low wind speed conditions (1.5 m/s).

Something wrong in your SAR data, too large contamination by SAR noise? If not, how do you explain the observations? Would be nice to see RVV/HH vs. mpf plot for the T4 case. How RVV/HH vs. mpf in T4 case changes if you set a lower signal-to-noise ratio for s0 data, for example 5dB, i.e. only those RVV/HH datapoints with SNR larger than 5 dB for s0HH and s0VV are accepted? I think you need to investigate further your T4 data.

I am should have presented this observation in my first review, sorry about this…but I noticed this now.

These are very interesting comments and questions. First of all, due to the low wind speed in T4 (0.6 m/s), we would expect specular reflection from the ponds. Hence, the backscatter signal from the ponds is most probably weak or lacking, which is mirrored by the decrease in intensity with increasing melt pond fraction. The low SNR could also contribute to mask out a weak melt pond signal. Co-polarisation ratio is expected to increase with melt pond fraction due to the difference in relative permittivity between sea ice and water. However, if there is no signal from the melt ponds, co-polarisation ratio is only representing the sea ice, and is not anymore related to melt pond fraction. The decrease of co-polarisation ratio with melt pond fraction in T4 is, in our opinion, only reflecting slightly different sea ice types/sea ice surface roughnesses surrounding the melt ponds in areas with high and low melt pond fraction. The correlation value of 0.31 is also very low (0.3 would reflect a p-value larger than 0.05), indicating a weak relation. This could be observed in the figure attached below, showing a scatter plot of co-polarisation ratio vs. $f_{MP}$.

Scharien et al (2014a) found, as the reviwer states, a strong positive relationship between co-pol ratio and $f_{MP}$ during low wind speeds (1.1 m/s). There are several differences between our study and Scharien et al (2014a), that could all contribute to the different findings. We had an even lower wind speed during acquisition of T4 (0.6 m/s) and a lower SNR, which could both make a large difference. The study of Scharien was also performed on smooth land-fast sea ice, a larger degree of smoothing was utilized, and the incidence angle was slightly higher. To directly compare the two studies is therefore difficult, and demonstrate the need of investigating a variety of conditions, wind speeds and incidence angles in future studies.

The relation between intensity and $f_{MP}$ found in Yackel and Barber (2000) is very weak (R^2=0.01!). Again, slightly higher wind speed (1.5 m/s), different SNR, different incidence angle, and different sea ice type could all contribute to difference in findings from our study.

The SNR ratio of T4 is low, and as mentioned above, this is absolutely a concern in our study. However, the relation displayed between intensity and $f_{MP}$ is statistically significant (p-value <0.05). Maybe a higher SNR ratio could have given even better results? A further investigation of the T4 data, and a general study on noisefloor effect on sea ice studies with TerraSAR-X would be very interesting, and could be a task of a future study.

[Figure]

To clarify the above-mentioned issues for the reader, the following changes have been made in the manuscript:

- The following sentences were rephrased (Discussion P13L480-492, P15L492-505): "Scene T4 represents a low wind speed situation ($U_2$=0.6 m/s), and our results indicate specular, or close to specular reflection from the melt ponds in this case. The weak melt pond backscatter, combined with a low SNR, hamper the use of difference in polarimetric properties between sea ice and melt ponds for melt pond fraction retrieval. The weak correlation seen between $R_{VV/HH}$ and $f_{MP}$ in Table 4 is most probably reflecting slightly different sea ice surface types surrounding the ponds in areas with low and high melt pond fraction, rather than different polarimetric signatures between melt ponds and sea ice. The low correlation observed between $R_{VV/HH}$ and $f_{MP}$ in this low wind case is in agreement with findings in Scharien et al (2012) and (2014b), while Scharien et al. (2014a) found a relation between $R_{VV/HH}$ and $f_{MP}$ even at low wind speeds ($U_{10}$=1.1 m/s). Different wind speeds, incidence angle and sea ice types could all contribute to the deviating findings. The lack of backscatter from the melt pond surfaces compared to the sea ice could potentially be used for $f_{MP}$ retrieval utilizing σ°,as the backscatter intensity becomes weaker with increasing $f_{MP}$."

Finally, Yackel and Barber (2000) speculated that MPF may be more closely related to the albedo than to melt pond fraction due to the fact that albedo results from the integration of all surface types (snow, saturated snow, melt ponds) which contribute to the measured MPF. What's the authors' view on this; would it be better to investigate the relationship between SAR data and albedo than SAR and melt pond fraction? Please, discuss this in Introduction Section.
This question might have a typing error, and we find it slightly unclear. We interpret the question to ask whether estimated melt pond fraction should be compared to albedo instead of observed melt

pond fraction. Albedo refers to the average reflection of waves in the visible range of the microwave spectrum. As SAR uses microwaves to evaluate the sea ice surface, we find it more credible to utilize differences in the microwave signature between melt ponds and sea ice, or methods that employ statistical features describing fractional mixtures of surfaces.

In our study, albedo is not measured, and would have to be estimated by upscaling from melt pond fraction measurements and in situ measured albedo values of different surface types. This method would inevitably introduce additional uncertainty to the results (see f.ex. Divine et al., 2015 for estimates made for the study area), and is therefore not advisable for our data set. The relationship between albedo and polarimetric features will therefore not be discussed in detail in our manuscript.

My idea was that as backscatter measured with SAR, if pixel size is not too small, is a mixture of backscatter from meltponds, bare ice, snow covered ice, then the backscatter could be more related to albedo than melt pond fraction. Albedo is naturally derivative of mpf. This is what Yackel and Barber (2000) speculated, see page 22068, left column. But you can leave this out the paper.

**Specific comments**
**Abstract**
"The regression fits gave good estimates of mean melt pond fraction for the full satellite scenes, deviating with less than 4% from the airborne retrieved melt 15 pond fractions in the investigated area."
Mention shortly the airborne data used in the melt pond fraction estimation.
This comment is slightly unclear for us; we have tried to rephrase the sentence to clarify the origin of the airborne retrieved melt pond fraction:
Sentence rephrased (Abstract P1L11-14, P1L11-15): "The regression fits gave good estimates of mean melt pond fraction for the full satellite scenes, with less than 4% deviation from a similar statistics derived from analysis of low altitude imagery captured during helicopter ice-survey flights in the study area."

"A smoothing window of 51x51 pixels gave the best reproduction of the width of the melt pond fraction distribution."
Give also pixel size of SAR images, only window size in pixels does not tell geometric size on the area.
The pixel spacing of each of the SAR satellite scenes are given in Table 1, and as can be seen in the table, they vary slightly from scene to scene. Hence a 51 x51 pixel window would have different geometrical sizes in each satellite scene. We find it cumbersome to include this in the abstract, but have included geometrical window sizes in the results, as commented on in the answer to the editor's comments.

"optical satellites" better "optical imagery"
The sentence where this term occurred was removed due to the editor's comments.

**1 Introduction**
l. 36: "Formation and evolution of melt ponds are poorly represented in sea ice models, potentially contributing to an underestimation of the observed sea ice extent reduction in model projections (Flocco et al., 2012; Holland et al., 2012; Flocco et al., 2015)."
Possible to explain shortly the reason for this underestimation? If not, then there are these three refs for the readers.
We would like to keep this paragraph brief, and will therefore leave it up to the readers to go in depth on this.

l. 42: "Several algorithms have been developed for retrieval of melt pond fraction…"
fMP defined before, could be used instead of "melt pond fraction". Check also whole text for the same issue.

We have now changed melt pond fraction to its abbreviation several places in the manuscript.

l. 104: "TerraSAR-X offers very high resolution multi-polarimetric data,…"
What do you mean by multi-polarimetric data? You have dual-polarisation TSX SAR images.
Multi-polarimetric data is a collective term for data with more than one polarization. To be more precise, we have changed the sentence:
- The following sentence was rephrased (Introduction P4L99-100, P4L102-103): "TerraSAR-X offers very high resolution dual-polarimetric images,…"

You could add at the end of Introduction Section short overview of the content paper Sections. Like "First, Section 2 presents …"
As all chapters have self-explanatory titles, and the last paragraph of the introduction already explains the presented work in conceptual terms, we have decided to avoid this extra text in the manuscript.

**2 Melt ponds in SAR imagery**
l. 118: "During very calm conditions, the SAR signal of melt ponds is mainly specular."
Maybe better "scattering from melt ponds is mainly specular"
We agree in this and have done the following change:
- Sentence rephrased (Melt ponds in SAR imager P4L112-113, P4L116-117): "During very calm conditions, the scattering from melt ponds is mainly specular."

p. 3, l. 127: "The coverage of melt ponds varies during the melt season, starting out with a high fractional cover, and reducing as the ponds drains. At the end of the melt season, the melt ponds refreezes."
I think melt pond fraction first increases, has a maximum level, and then decreases, ending up with remaining ponds refreezing, see e.g Fig. 6 in Rösel, A., Kaleschke, L., and Birnbaum, G.: Melt ponds on Arctic sea ice determined from MODIS satellite data using an artificial neural network, The Cryosphere, 6, 431–446, doi:10.5194/tc-6-431-2012, 2012.
We agree that the sentences were slightly unprecise, and have done the following changes:
- Sentences rephrased (Melt ponds in SAR imagery P4L121-123, P4L125-129): "The coverage of melt ponds varies during the melt season, increasing from melt onset until it reaches a maximum level, and then gradually reducing as the ponds starts to drain \citep{Barber2001}. At the end of the melt season, the remaining melt ponds refreeze."

l. 184: "The sea ice surface roughness was found to high to fill the criterion…"
'too high'
The spelling error was corrected.

l. 201: "Entropy (H) is a part of the H=A= polarimetric decomposition, based on the eigenvectors and eigenvalues of T , describing SAR scattering mechanisms."
This sentence should have a reference to the polarimetric decomposition.
A reference to Cloude and Pottier (1997) was added to the sentence.

**3 Methods**
Figure 2 in grayscales could show better sigma0 variation.
The figure is now in grayscale.

**4 Results**
l. 375: "and values in parentheses show results after before NESZ subtraction of the signal"
…NESZ subtraction in the SAR imagery…

The sentence was removed as information about the results before NESZ subtraction have been removed from the paper.

Figure 4: gray and blue dots does not separate well; very long minus sign on -0.49
The color of the dots have been slightly modified, and the minus sign is removed (it should be +)

Figure 6 and 10: what is the spatial size of the images in km?
The spatial size of the images is 0.3 x 1.1 km, this has been added in the figure captions.

**5 Discussion**
l. 558 "From the helicopter images, some of the very low RVV/HH values observed at the investigated floe in scene T3 were from slightly deformed areas, possibly explaining the negative ratios. However, no general trend in low RVV/HH values in deformed areas was found in our study."
Paper: Sea ice type and open water discrimination using dual co-polarized C-band SAR, T Geldsetzer and J J Yackel, Canadian Journal of Remote Sensing Vol. 35 , Iss. 1,2009 4
may have intetesting results in the context of your paper. Average RVV/HH larger than 0 dB for various ice types, but <0 dB observations also exist.

In Figure 4 large amount of negative RVV/HH values, not saying they are wrong, but kinda puzzling…calibration error between TSX HH and VV channels? If so, I guess there is nothing you can do the correct it.
We agree that negative co-polarisation ratio values seem a bit odd. As we have written in the manuscript, the negative values might relate to multiple scattering from deformed areas, but there could be other reasons as well, for instance those described by Geldsetzer et al (2009). Scharien et al (2014b) also reports negative co-polarisation ratio values (see Figure 8) both for bare ice and ponds. We cannot exclude calibration errors between the HH and VV channels, but as you write, this is hard to verify and correct. T3 is also well above the noise floor, so low SNR cannot explain the negative values. We have decided to include a reference to Geldsetzer and Scharien in the text, slightly expanding the above referred sentences:

- Sentences and references added (Discussion P16L522-524, P16L535-537): "Other small scale surface scattering processes could also have caused low $R_{VV/HH}$, negative values have also been reported in other FYI studies, e.g., Geldsetzer and Yackel (2009) and Scharien et al. (2014b)."

**6 Conclusions**
l. 632: "At very low wind speeds (0:6 m/s), the backscatter signal from the melt ponds became too low for fMP retrieval based on polarimetric features. In that case, s0VV was found suitable for fMP estimation."
If SNR is too low for polarimetric features, then why it is not too low for s0? Again, the T4 data may be too continimated with SAR noise for mpf retrieval studies.
The above-referred sentence discusses the weak or lacking backscatter from the melt ponds in T4, which in our opinion is related to low wind speed at the acquisition time. The low SNR might also contribute to mask the melt pond signature in the SAR images. The consequence of a lacking signal from the melt ponds is that the difference in polarimetric properties between melt ponds and sea ice cannot be used for melt pond detection in this case. However, the lack of backscatter from the melt ponds would lead to a lower backscatter intensity in the areas with high melt pond fraction. This could explain the relation between sigma VV for melt pond fraction in the case of T4. We have tried to clarify this in the manuscript:

- The following sentences were rephrased (Conclusion P18L595-597, P18L613-615): "
[revised manuscript text omitted]